# Customizable wave tailoring nonlinear materials enabled by bilevel inverse design

Brianna MacNider [1,4], Haning Xiu [1,4], Caglar Tamur [2], Kai Qian [1], Ian Frankel[1], Maya Brandy[1], Hyunsun Alicia Kim[2,3] & Nicholas Boechler [1,3] ✉

Passive wave transformation via nonlinearity is ubiquitous in settings from acoustics to optics and electromagnetics. It is well known that different nonlinearities yield different effects on propagating signals, which raises the question of "what precise nonlinearity is the best for a given wave tailoring application?" In this work, considering a one-dimensional spring-mass chain connected by polynomial springs (a variant of the Fermi-Pasta-Ulam-Tsingou system), we introduce a bilevel inverse design method which couples the shape optimization of structures for tailored constitutive responses with reduced-order nonlinear dynamical inverse design. We apply it to two qualitatively distinct problems—minimization of peak transmitted kinetic energy from impact, and pulse shape transformation—demonstrating our method's breadth of applicability. For the impact problem, we obtain two fundamental insights. First, small differences in nonlinearity can drastically change the dynamic response of the system, from severely under- to outperforming a comparative linear system. Second, the oft-used strategy of impact mitigation via "energy locking" bistability can be significantly outperformed by our optimal nonlinearity. We validate this case with impact experiments and find excellent agreement. This study establishes a framework for broader passive nonlinear mechanical wave tailoring material design, with applications to computing, signal processing, shock mitigation, and autonomous materials.

The passive transformation of waves via nonlinear material response is widely used in physical settings ranging from acoustics[1] to optics[2-4] and electromagnetics[5,6]. Applications include areas such as efficient information transfer[7,8], computing and logic[9,10], energy conversion[11], imaging[12], encryption[13], impact and vibration mitigation[14,15], and rapid shape change[16]. Within these contexts, in contrast to active control, passivity is of particular importance for responding fast to stimuli, and nonlinearity enormously expands signal transformability compared to linear systems due to the breaking of superposition. Indeed, it is well known that different types of nonlinearity yield vastly and qualitatively different effects on propagating signals[17], which raises the question of "what precise nonlinearity is the best for a given wave tailoring application?" This

question has largely remained in the regime of simulation and theory, as, until recently, it has not been possible to freely realize any optimal nonlinear constitutive law in practice. The field of mechanics has come furthest towards this goal, by introducing complex, sub-wavelength, geometric motifs to create "mesostructured" nonlinear materials[1], however the tunability was, until recently, coarse and limited around a handful of known nonlinear mechanisms. For instance, a few broad classes of nonlinearity that have seen tailorability for wave manipulation include contact nonlinearities[18], tensegrity structures[19], and bistable beam arrays[20], among others[1,21].

Recent progress has enabled a, thus far unique-to-mechanics, capacity to create materials with on-demand quasi-static nonlinear

[1]Department of Mechanical and Aerospace Engineering, University of California, San Diego, La Jolla, CA, USA. [2]Department of Structural Engineering, University of California, San Diego, La Jolla, CA, USA. [3]Program in Materials Science and Engineering, University of California, San Diego, La Jolla, CA, USA. [4]These authors contributed equally: Brianna MacNider, Haning Xiu. ✉e-mail: nboechler@ucsd.edu

properties via shape and structural optimization[22–29]. This has included several approaches, including gradient based topology optimization in pursuit of tailoring the entirety of a nonlinear force-displacement curve[22–24] as well as the incorporation of machine learning (ML) algorithms in an attempt to traverse the design space and speed up predictions of mechanical behavior[25,27–30]. However, such methods alone cannot identify material designs for optimal system-level nonlinear wave tailoring performance. Prior studies of optimal nonlinear dynamic material behavior have tailored heterogeneity with fixed nonlinearity[31] or dynamic behavior where the characteristic wavelengths are on par with or greater than the system size (and thus the response is not "wave-dominated") and the tailoring was confined to broad metrics like "area under the curve"[32,33] or "plateau-like" behavior[34]. The role of waves is of particular importance, as allowing for spatiotemporal evolution in nonlinear systems leads to unique emergent phenomena such as solitons[35]. The role of precisely engineered nonlinearities is further important for wave propagation in that seemingly subtle differences in nonlinearity yield *qualitatively* different dynamical behavior. For one example, consider a material with polynomial nonlinearity and all positive coefficients, resulting in a "stiffening" nonlinearity: just small changes in the ratio of coefficients dictate whether or not the system experiences modulational instability[36,37]. Connecting the inverse design of nonlinear wave response to the quasi-static design of nonlinear constitutive response induced by mesostructure geometry is a significant, and hitherto unsurmounted challenge. If trying to directly extend quasi-static geometric design algorithms based on finite element method (FEM) simulation[22–29] the challenge becomes evident, in that one would need to take the same design variables, copy the geometry over many unit cells, and simulate the entire system in time at high temporal resolution (due to nonlinear generation of high frequency content), and wrap that in an automated design loop—resulting in a task of extreme computational expense.

In this work, we introduce a method to create customizable wave tailoring materials via nonlinear bilevel inverse design. Namely, we optimize for the emergent dynamic response of a mesostructured material in the form of a one-dimensional (1D) spring-mass chain connected by polynomial coefficients. To do this, we use a reduced order, discrete element model (DEM) simulation to identify an optimal nonlinear constitutive law for the given performance metric, and couple this to a unit-cell-scale, geometrically-nonlinear, free-form, shape optimization algorithm which designs a physical system that achieves the nonlinear constitutive property identified by the DEM (outlined in Fig. 1). Unlike some prior computational quasi-static nonlinear mechanical design strategies[23,25,26,28], we do not use simplified or reduced order models for our underlying mesostructure design, which enables a broader design space and access to highly precise tailoring of nonlinear responses[24]. We note that this chain is a variant of the celebrated Fermi-Pasta-Ulam-Tsingou (FPUT) model[38], whose initial study is widely regarded as responsible for the birth of experimental mathematics[39]. The FPUT system has also been shown equatable to nonlinear continuum models such as the Kortweg-de Vries (KdV)[35] and the nonlinear Schrödinger equation (for the case of envelope solitons in diatomic systems[40]), and formed the foundation for extensions into higher dimensions[17]. Considering the latter, as part of our work herein, we illustrate extensions of our optimized unit-cells to two- and three-dimensional (2- and 3D) analogs (see Supplementary Information Note 1). In addition to the introduction of this method, we apply it to two qualitatively distinct problems—minimization of peak transmitted kinetic energy in response to an impact, and pulse shape transformation (inversion of an applied boundary displacement signal at the other end of the chain)—demonstrating the potential breadth of applicability of our method. We highlight also that in both cases, we conduct a comparison between the linear and nonlinear response. We assert that this comparison is particularly important from a fundamental perspective, as it isolates the role of nonlinearity from other linear wave manipulating effects such as dispersion, dissipation, and heterogeneity.

Focusing on the problem of minimizing peak kinetic energy transmitted via waves in response to impact, we first note that because impact is inherently a broadband excitation, prior linear mesostructured material (or "metamaterial") strategies that leverage bandgaps[41] have been shown to have limited efficacy (e.g., requiring gradient or disordered material strategies that increase bandwidth at the cost of attenuation[42]). In favor of this, several nonlinear mesostructured material motifs have been realized and studied in the context of impact, leveraging nonlinearities such as the aforementioned contact (tensionless and stiffening in compression[14]), tensegrity

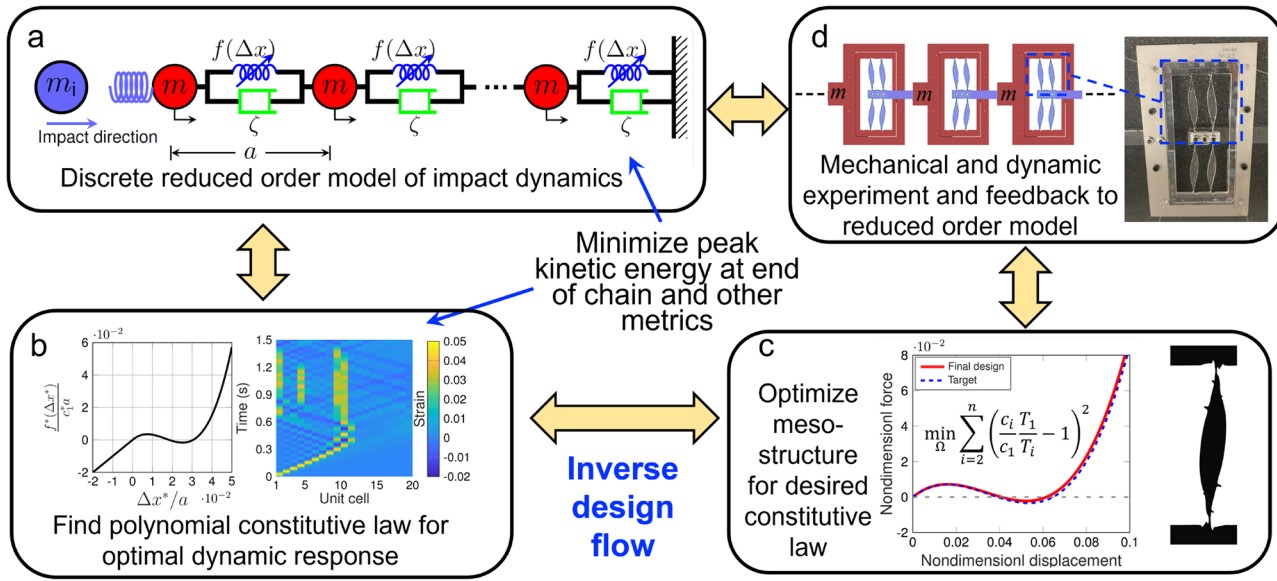

**Fig. 1 | Overview of the bilevel design flow. a** Discrete element model (DEM) simulation of the entire system dynamics with boundary configured for the impact problem. **b** Identification of the optimal nonlinear constitutive law. **c** Shape optimization of a mesostructure to match the identified nonlinear constitutive law. **d** Mechanical and dynamic experimental system characterization. Source data are provided as a Source Data file.

(stiffening in tension and tunably softening or stiffening in compression[19]), and compressed beams (bistability and snap-through in compression[43]), as well as o-rings (tensionless with double-power law response in compression[44]) and origami (softening in compression and stiffening in tension[45]). Herein, using our method, we obtain the following two fundamental insights within this area. The first, is that very small differences in the nonlinear spring response (without qualitative difference) can drastically change the response of the system (changing the nonlinear coefficients by less than 14% results in an over 3000% change in performance) in such dynamical settings, from severely under- to outperforming a comparative linear system. To our understanding, this degree of sensitivity was hitherto unknown in the context of impact mitigation. The second, is that the often used design strategy of impact mitigation via "energy locking" bistability[15,43] can be greatly outperformed by our identified optimal snap-through nonlinearity (by over a factor of three times). We then choose a second high performing identified nonlinear constitutive response (which is more amenable to our experimental capabilities) and demonstrate the full inverse design of a superior impact mitigation system, from identification of an ideal nonlinearity, to the design of the unit cell geometry, and experimental validation of the performance. While we focus the majority of the manuscript on this impact mitigation problem, as demonstrated by our application of this method to the aforementioned two qualitatively different objectives, we believe our method will have wide future applicability, including to the previously referenced array of nonlinear acoustic, phononic, and mechanical wave transformation applications[1,7–13].

## Results

### Reduced order discrete element model

The DEM (Fig. 1a) simulates the dynamical response of a chain with $N$ unit cells and $a$ unit cell length. We represent the chain as lumped masses of mass $m$, interconnected by massless nonlinear springs and inter-site linear dampers in parallel, emulating the behavior of a viscoelastic material. The nonlinear spring consists of an up-to-third-order polynomial, where the linear stiffness remains fixed at $c_1^*$. The non-dimensional nonlinear spring force is expressed as

$$f(\Delta x) = \Delta x + c_2(\Delta x)^2 + c_3(\Delta x)^3, \qquad (1)$$

where $\Delta x = \Delta x^*/a$ is the dimensionless spring stretch (with $\Delta x^*$ the spring stretch defined such that elongation is positive), and $c_2$ and $c_3$ are dimensionless nonlinear coefficients of the second and third order-terms (with $c_n = c_n^* a^{n-1}/c_1^*$). We choose to describe the nonlinear springs as up-to-third-order polynomials due to the flexibility of this representation, namely, the ease with which they can represent a wide qualitative range of nonlinearities and the ease which polynomials lend to accurate dynamical simulation (as opposed to non-differentiable, e.g., piecewise continuous functions). Along these lines, we found that inclusion of up to fifth order polynomial terms resulted in minimal performance improvements compared to the third order representation for our primary case study of peak transmitted kinetic energy minimization (see Supplementary Information Note 2). This results in non-dimensionalized equations of motion for our chain

$$\ddot{x}_i - (x_{i+1} - x_i) + c_2(x_{i+1} - x_i)^2 - c_3(x_{i+1} - x_i)^3 + (x_i - x_{i-1})$$
$$- c_2(x_i - x_{i-1})^2 + c_3(x_i - x_{i-1})^3 + 2\zeta(-\dot{x}_{i+1} + 2\dot{x}_i - \dot{x}_{i-1}) = 0, \qquad (2)$$

where $\zeta$ is the inter-site damping ratio, $x_i$ is the dimensionless displacement of the $i$th particle from its rest position, overdots represent the derivative with respect to nondimensional time, and all variables and parameters are normalized by combinations of $a$, $m$, and/or $c_1^*$, as is described in Supplementary Information Note 3. We note that while the masses are illustrated as $m$ in Fig. 1a, the nondimensional

mass of each particle remains one as in Eq. (2). A fixed boundary is applied on the right. The simulated dynamical response is acquired through the numerical integration of Eq. (2) via a Runge-Kutta algorithm. The non-dimensionalization of all variables and full details concerning the equations of motion are described in Supplementary Information Note 3.

### Optimization of nonlinear constitutive law based on dynamical response: minimizing peak kinetic energy transmission as a case study

In this section, we describe the identification of an optimal nonlinear constitutive law for the case study of minimizing transmitted peak kinetic energy in response to an impact. Specifically, as shown in Fig. 1a, we simulate the impact of a rigid, variable mass and velocity rigid "impactor" incident on the left end of the chain. The model also incorporates a contact spring designed to facilitate the smooth contact and controlled release of the impactor during initial impact and rebound, respectively. We first consider a single impact condition ($M/M_0 = 0.05$ and $V/V_0 = 1$), where $M_0$ is half the mass of the chain, $V_0$ is the linear sound speed, and $M$ and $V$ are the dimensional impactor mass and velocity, respectively. The material is composed of 20 particles and $\zeta = 0.01$. Our control parameters (i.e., design variables for optimization) are the nonlinear coefficients of the springs $c_2$ and $c_3$, which we vary in the aim of minimize the maximum kinetic energy experienced at the end of the material ($KE_{non}$) normalized by that of a linear system ($KE_{lin}$) which has all of the same properties except $c_2 = c_3 = 0$ (we herafter refer to this ratio as the "KE ratio", where smaller numbers equate to better performance of the nonlinear system). Given dimensional particle displacement from its rest position $x_i^*$, kinetic energy of the $i$th particle is defined as $m(dx_i^*/dt^*)^2/2$, where $t^*$ is dimensional time, such that the analogous dimensionless kinetic energy is $\dot{x}_i^2/2$.

Before searching for optimal nonlinear constitutive responses with our DEM, we set several bounds. First, for simplicity, we set the tension response to purely linear. In Supplementary Information Note 4, we show that the inclusion of nonlinearity in tension has little effect on the identified optima, which is as expected, due to the compressive nature of the impact event simulated (the identified optimum has a maximum compressive strain over five times greater than the maximum tensile strain). Second, we confine the unit cell strain to 1 in compression, and set $c_3 > 0$ for simplicity. Third, we restrict our search range for nonlinear coefficients $c_2$ and $c_3$ to ensure positive strain energy throughout the entire compression range. We note that keeping the linear stiffness constant, we exclude essential nonlinearities[14]. By examining the polynomial's properties within this range, we classify the quasi-static response into three distinct zones, "bistability", "monotonic increase", and "local maximum", as shown in Fig. 2a. Bistability (magenta area) denotes the existence of both a local maximum and minimum other than the boundaries (the local minimum does not need to fall below zero). Monotonic increase (blue area) denotes the absence of extrema. Local maximum (green area) signifies the presence of a local maximum (no local minimum existed) within the range of the length of one unit cell. For a monotonic increase of $f(\Delta x)$, the condition $f'(\Delta x) \geq 0$ must be satisfied, or the local maximum of $f(\Delta x)$ should occur at $\Delta x \geq 1$. To ensure bistability, both local maximum and minimum of $f(\Delta x)$ are set to be located within $\Delta x \in (0, 1)$. For a local maximum property to be exhibited, we must have $0 < \Delta x_1 < 1$ and $\Delta x_2 > 1$, where $\Delta x_1$ and $\Delta x_2$ are the roots of $f'(\Delta x) = 0$. More details concerning these zones are given in Supplementary Information Note 5.

Using a gradient-based optimization algorithm (see Methods), the best performance within the described context is found to correspond to a nondimensional spring force $f(\Delta x) = \Delta x - 5.88\Delta x^2 + 9.65\Delta x^3$, where $\Delta x$ is the spring extension and positive $f$ denotes compression. To visualize the design space, we also sweep the nonlinear coefficients

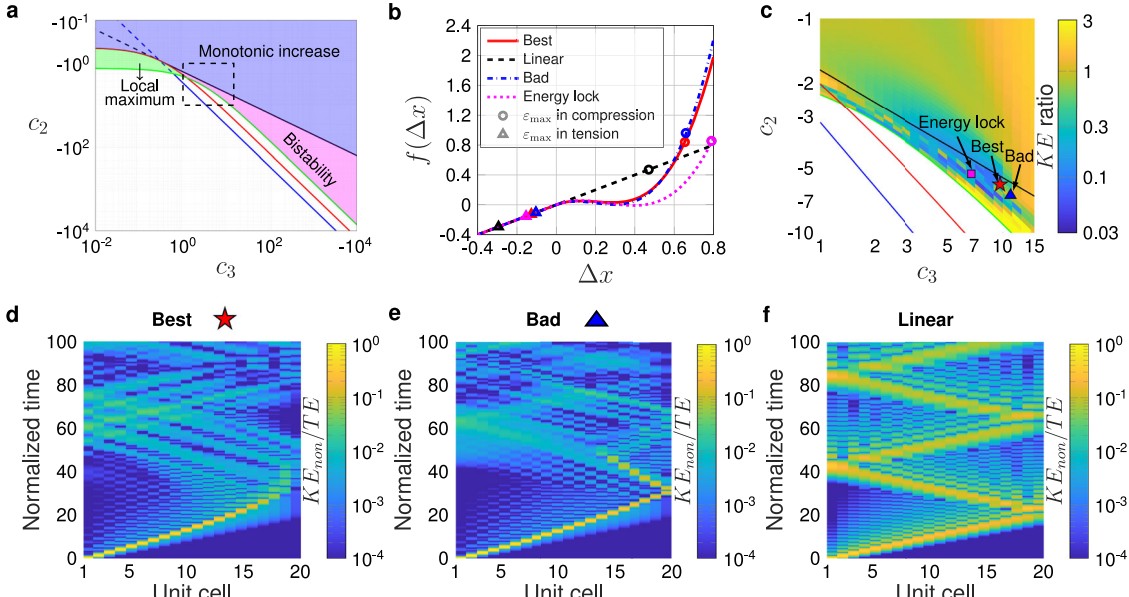

**Fig. 2 | Identification of optimal nonlinear constitutive response via DEM simulation for a single impact condition for the case of peak transmitted kinetic energy minimization. a** Feasible solutions of nonlinear spring coefficients $c_2$ and $c_3$. The black line represents $c_2 = -\sqrt{3c_3}$, the red line indicates $c_2 = -(1 + 3c_3)/2$, the blue line is $c_2 = -3c_3$, and the green line is the zero strain energy throughout the whole range, $c_2 = -3/2 - 3c_3/4$. **b** Non-dimensional force-extension relationship of the best performing nonlinear spring ($f(\Delta x) = \Delta x + 5.88\Delta x^2 + 9.65\Delta x^3$) along with an example of a nearby underperforming (bad) nonlinear spring of $f(\Delta x) = \Delta x - 6.6\Delta x^2 + 11\Delta x^3$ (blue) and an energy locking bistable spring of $f(\Delta x) = \Delta x - 5.26\Delta x^2 + 6.75\Delta x^3$ (magenta). These three cases result in KE ratios of 0.0398

(best), 1.2257 (bad), and 0.1448 (energy locking bistable). Circles and triangles indicate the maximum compressive and tensile strain, respectively. **c** Ratio of maximum kinetic energy of the nonlinear spring to the one of a linear spring at the last particle as a function of nonlinear spring coefficients for the impact condition of $M/M_0 = 0.05$ and $V/V_0 = 1$, with $\zeta = 0.01$. The lines from (**a**) are overlaid, the star marker denotes the point of best performance, the triangle indicates the nearby case, and the square represents the bistable case. Normalized kinetic energy of the (**d**) best performing nonlinear, (**e**) underperforming (bad) nonlinear, and **f** linear material. Source data are provided as a Source Data file.

directly and plot the KE ratio, as is shown in Fig. 2c. The optimal spring is plotted in Fig. 2b, and is denoted by the star marker in Fig. 2c, alongside a nearby underperforming spring (triangle) and a conventional "energy locking" bistable spring[15,43] (square). These three cases result in KE ratios of 0.0398 (best), 1.2257 (bad), and 0.1448 (energy locking bistable), respectively. Given the underperforming (bad) spring has a nondimensional spring force of $f(\Delta x) = \Delta x - 6.6\Delta x^2 + 11\Delta x^3$, we note the extreme sensitivity for such a dynamic problem. Change by less than 14% in the nonlinear coefficients (calculated as $\max[1 - c_{n,bad}/c_{n,best}]$) results in over 3000% improvement (calculated as KE ratio$_{bad}$/KE ratio$_{best}$) in performance between the optimum and the underperforming spring (where the "best" and "bad" subscripts denote their respective cases).

A further important point of note about our identified optimum is that it is not of the form one would expect based on the conventional design approach for bistable energy absorption at lower rates[43], where net positive energy is locked into strain energy when snapping from the undeformed state to its second stable equilibrium (i.e., where for energy locking, the area under the curve from $\Delta x = 0$ to the unstable equilibrium point is greater than the area under the curve from the unstable equilibrium to the second stable equilibrium point). In the case of our optimum, the response exhibits snap-through, but is neither bistable nor satisfies the more restrictive case of energy locking. Using this qualitatively different nonlinearity, our optimum outperforms the energy locking bistable case by over a factor of three (300% change via the above metric).

We next seek to understand why our optimum performs better than the other cases. Spatiotemporal responses of kinetic energy of the optimal nonlinear, underperforming nonlinear, and linear chain are shown in Fig. 2d–f, respectively. Most notably, while all materials in Fig. 2d–f see a pulse of energy propagating across the material, the pulse in the best performing case (Fig. 2d) appears to stop before the

end of the chain. Via the metric of minimizing peak kinetic energy at the end of the chain, it is clear why this case performs better. We next make several observations about the spatiotemporal responses. First, the traveling pulses in the two nonlinear cases (Fig. 2d, e) are more localized than that of the linear case (Fig. 2f). This is to be expected, due to the known formation of solitary waves (which localize via a balance of nonlinear and dispersive effects) in systems with qualitatively similar nonlinearities (snap-through)[46]. Indeed, in ref. 46, they show that these solitary waves take the form of "boomerons", where the wave arrest (and in their case reversal of direction) arises without any dissipation, and is suggested to be "a consequence of the intriguing interaction between the localized phenomenon and the trail of nonlinear waves". In the Supplementary Information Note 6 we show spatial profiles of the pulses in our systems at several times to further visualize the observed localization phenomena.

However, we suggest that the boomeron-related wave arrest found in conservative systems is not the only mechanism contributing to the identified optimum performance. In Fig. 3a, b, we show the 2D Fourier transforms of the normalized velocity of the optimal nonlinear (A) and linear (B) systems. In the linear system (Fig. 3b), the energy distribution can be seen to follow the expected dispersion of a monoatomic chain (a single "acoustic" branch, followed by stop band above a cutoff frequency[47]). In Fig. 3a, due to frequency conversion induced by the system's nonlinearity, the energy is spread out to a much broader frequency range. This spreading has two effects. First, it converts some of the wave energy into the non-propagating stop band region, and second, it generates higher frequencies which more strongly activate energy loss via the inter-site damper (due to its proportionality with velocity). This latter effect can be seen in Fig. 3c, wherein the total energy of the entire chain decreases more quickly in the two nonlinear systems (both the best and the bad cases) than the linear system. This suggests that the ideal use of the underlying

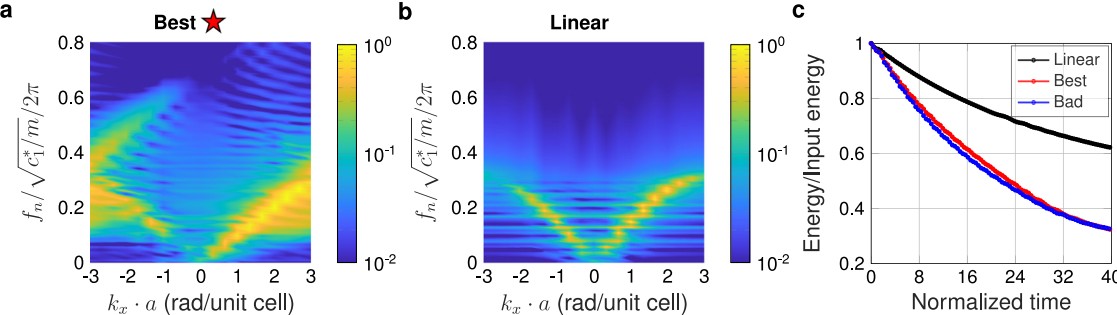

**Fig. 3 | Contribution of the interaction of dissipation and nonlinearity to the minimization of transmitted peak kinetic energy.** Fourier transforms (dimensionless frequency vs. wavenumber) of the spatiotemporal normalized velocity of the optimal nonlinear (**a**) and linear (**b**) systems (corresponding to the spatiotemporal response of Fig. 2d and f, respectively. **c** Time evolution of total energy in the system, normalized by initial total energy (impactor kinetic energy) for three cases mentioned above. Source data are provided as a Source Data file.

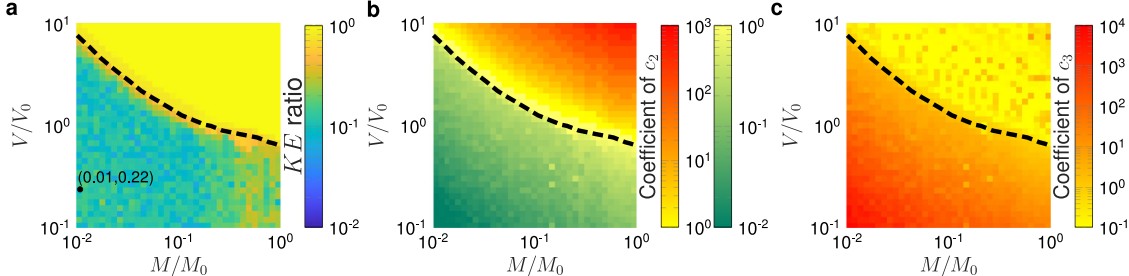

**Fig. 4 | Identification of optimal nonlinear coefficients via DEM simulation for varied impactor conditions for the case of peak transmitted kinetic energy minimization. a** Optimal kinetic energy ratio (*KE* ratio) as a function of impact conditions and corresponding nonlinear spring coefficients **b** $c_2$ and **c** $c_3$ for a material of $N = 20$ and $\zeta = 0.005$. The color bar in (**a**) is saturated at a *KE* ratio of unity. The dashed black line denotes the onset of self contact ($\Delta x < -1$) in the linear system. The dot in (**a**) denotes the high performing case used for mesostructure design. Source data are provided as a Source Data file.

mechanisms for our peak transmitted kinetic energy minimization metric would thus be to decrease the total energy of the system via nonlinear-dissipation interplay, while simultaneously arresting the propagating pulse before the end of the chain via the aforementioned boomeron mechanism. Such interplay between dissipation, damping, and wave arrest is consistent with the recent observation in bistable nonlinear systems[15,48].

We next conduct an optimization (via parameter sweep) wherein we look for optimal nonlinear coefficients for varied impactor mass and velocities. The optimal *KE* ratios with respect to $M/M_0$ and $V/V_0$ and corresponding nonlinear spring parameters $c_2$ and $c_3$ can be seen in Fig. 4. In contrast to Fig. 2, we use lower damping ($\zeta = 0.005$), chosen to emulate that of the polycarbonate springs used in our experimental realization. Additional simulation results of *KE* ratios for increased damping and greater discreteness (more unit cells) are available in Supplementary Information Note 7 and indicate the potential for *KE* ratio < $10^{-2}$ in the latter case. The damping value used in Fig. 4 was chosen by measuring the low-amplitude resonance of a single connector and nonlinear spring unit (see Supplementary Information Note 8). Characterizing the damping at low amplitudes allows us to temporarily discard its interplay with the nonlinear spring, wherein the measured damping can be thought to stem from the intrinsic damping of the polycarbonate. Simulation cases of the nonlinear material where self-contact occurs are discarded from consideration of the optimal performance.

There is a clear boundary where the nonlinear chain does not outperform the linear, which is correlated with the occurrence of self-contact within the linear chain (dashed black line in Fig. 4). At impactor velocities and masses below this threshold, the nonlinear materials exhibit significantly enhanced mitigation effectiveness. As the impactor mass and velocity increase and cross over the dashed black line (orange to red area near the dashed line in Fig. 4a), the priority shifts to

preventing contact between unit cells, leading to a comparatively impaired energy-absorbing performance. In future studies, such self contact could be explored as a form of nonlinearity and a design feature instead of a constraint.

In the following, we choose a particular optimal solution (pair of coefficient ratios and impactor conditions), which is denoted by the black dot in Fig. 4a, for unit cell shape optimization and subsequent experimental validation. This solution was chosen in favor of that shown in Fig. 2, because it exhibited lower strains (easier convergence of the shape optimizer FEM kernel and avoidance of possible plasticity) and exhibited a relatively smooth local response to variations in impactor conditions, all of which make it more amenable to experimental implementation. The chosen solution is a nondimensional nonlinear mechanical response of the form $f(\Delta x) = \Delta x - 87\Delta x^2 + 1778\Delta x^3$ in compression, with an impact condition of $M/M_0 = 0.01$ and $V/V_0 = 0.22$. We note that this identified high-performance solution is bistable, in contrast to the snap-through case of Fig. 2.

## Shape optimization for desired effective nonlinear constitutive law and mechanical experimental validation

The constitutive response of a spring element is directly tied to both its constituent material and its geometry. Given that the response of the underlying constituent material is accounted for (e.g., neo-Hookean or Saint Venant-Kirchhoff), the geometry can be designed to tailor the effective constitutive response of the spring (force-displacement response simulated via the commercial FEM software COMSOL). Several example known mechanisms for achieving various broad classes of nonlinearity are highlighted in Fig. 5a. This is accomplished through coarse geometry adjustment—that is, only altering the angle, length, and thickness of the beam. The underlying mechanisms can be intuitively thought of as follows: when the thin beam-like spring element undergoes large deformation, if it deforms into a state where it is being

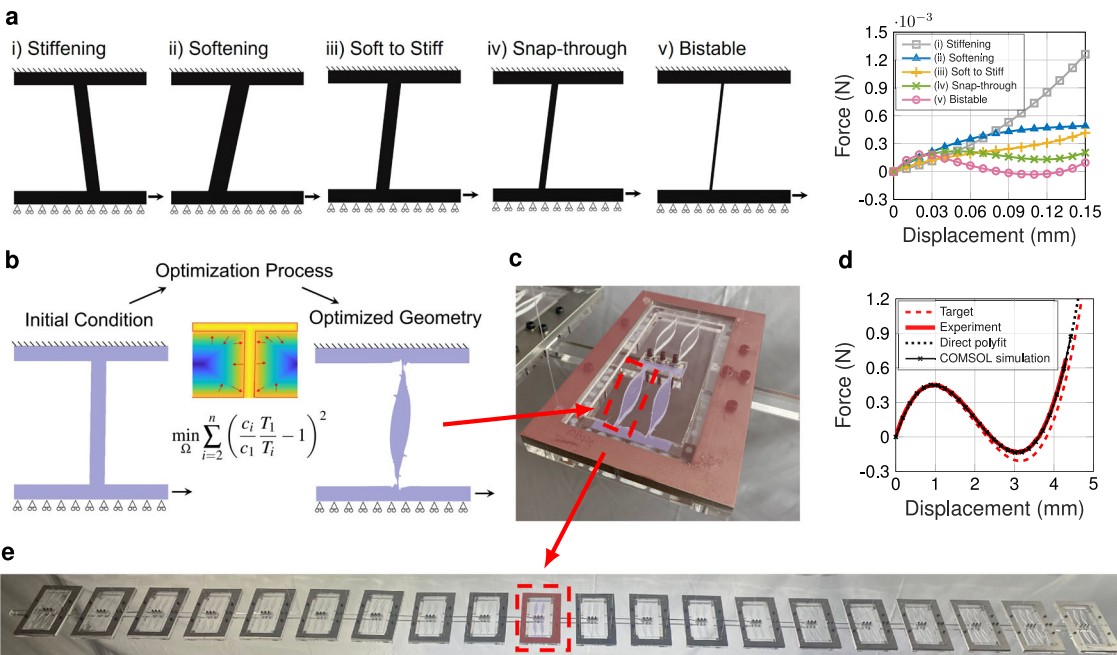

**Fig. 5 | Designing unit cells that leverage geometric nonlinearity to achieve DEM-identified desired effective constitutive laws, and experimental realization of the spring and chain. a** Example mechanisms which can be used to achieve various nonlinearities with geometry alone are shown, including the following nonlinearities: i) stiffening, ii) softening, iii) soft to stiff, iv) snap-through (but not bistable), and v) fully bistable. The plot on the right shows the nonlinear responses of these five example mechanisms (with the stiff to soft, snap-through, and bistable curves linearly scaled to be on the same scale as the thicker mechanisms).
**b**–**e** Optimized spring design and chain realization for the case of minimizing peak transmitted kinetic energy aimed to match the nonlinear coefficients identified in Fig. 4a. **b** Results of the shape optimization, with the initial condition and optimized design shown. **c** The fabricated polycarbonate unit cell, consisting of four optimized springs (as seen in (**b**)) and a rigid frame. **d** The quasi-static test of the unit cell shown in (**c**), compared against the target the behavior simulated via COMSOL FEM. **e** The full chain of 20 unit cells, hung from a frame. The chain is clamped to the left of the leftmost unit cell, imposing a zero displacement boundary condition. The impact occurs at the right end of the chain. Source data are provided as a Source Data file.

stretched axially, it will stiffen, while if it is deformed to where it is being loaded transversely, it will soften. If in the latter case it is deformed such that it undergoes axial compression against the boundaries, it tends towards negative stiffness (e.g., snap-through or bistability). While these coarse adjustments are adequate to achieve broad types of nonlinear behavior, they are inadequate to achieve the type of precision of nonlinear response needed in dynamical settings–e.g., the subtle difference between the optimal spring response (red line) and the underperforming (bad) spring response (dashed-dotted blue line) shown in Fig. 2.

Therefore, in order to find a spring geometry that gives the specific desired nonlinear response, a 2D shape optimization approach is taken, using a level set optimization method (as detailed further in the Methods and ref. 24). In summary, a third order polynomial is fit to the calculated force-displacement behavior of the structure, and the objective of the optimization problem is taken as the ratio of nonlinear to linear terms (e.g., as shown in Figs. 1c and 5b). The exact form of the objective function is

$$\min_{\Omega} \sum_{i=2}^{n} \left( \frac{c_i}{c_1} \frac{T_1}{T_i} - 1 \right)^2, \tag{3}$$

where $\Omega$ is the design domain (i.e., the range of values that all the design variables can take), $c_i$ and $c_1$ represent the current polynomial coefficients (nonlinear and linear, respectively), $T_i$ and $T_1$ represent the target coefficients (nonlinear and linear, respectively), and $n$ represents the total number of nonlinear polynomial coefficients. By taking the ratio of the polynomial terms, the nonlinearity of the structure is decoupled from the linear stiffness, allowing the optimizer more design freedom, leading to more robust convergence. This is a potentially subtle, but important point. If an optimizer is to seek a

match to an absolute force-displacement curve, there is the possibility that, given a chosen constituent material property, the optimizer may be forced for high absolute forces to increase the volume ratio to a degree that it does not leave the geometry with enough kinematic freedom to achieve the desired nonlinearity. Similarly, for very low absolute forces, the optimizer may be forced to pursue features below the discretization of the FEM simulation kernel, potentially disconnecting elements. As such, normalizing linear stiffness during the search (equivalent to having the constituent modulus as a design variable), is critical in achieving microstructures matching target nonlinearities with potentially far initial guesses. This normalized stiffness can similarly be rescaled in practical implementation via choice of constituent material or, in 2D, as we use herein, by adjusting the out-of-plane depth of the structure. In any case, to accelerate and ease the navigation of the design space, we select an initial condition for our optimization process, which displays qualitatively similar behavior to that desired (i.e., a slightly angled beam; see Supplementary Information Note 8).

The boundary conditions applied are depicted in Fig. 5b (fixed on top, roller on bottom, applied displacement on bottom 10% of the right boundary). When realizing our nonlinear chain, we employ comparatively rigid frames around the designed spring to mimic fixed boundary conditions (as can be seen in Figs. 1d and 5c) and rigid connectors between the springs to allow relative movement of the masses. With the addition of these components, we have a unit cell length, $a$, and a nonlinear spring design domain length that encompasses only a portion of this larger unit cell. We call this portion of $a$ the spring length, and denote it by $a_s$ (and, hereafter, the subscript $s$ is used to refer to parameters defined on the scale of the spring). Figure 5e highlights the difference between these two length scales. Because the DEM-identified polynomial constitutive law is expressed

as a function of strain, and the strain experienced by the spring across $a_s$ is different than that experienced by the entire unit cell across $a$ for a fixed applied displacement, the targeted polynomial is therefore scaled accordingly. We take the ratio of nonlinear terms as $R = (c_i \epsilon^i)/(c_1 \epsilon)$ and $R_s = (c_{s,i} \epsilon_s^i)/(c_{s,1} \epsilon_s)$ on the scales of $a$ and $a_s$, respectively, with $i$ representing the order (or power) of the term and $\epsilon$ representing the strain experienced on the corresponding length scale. In order for equivalent degrees of nonlinearity to be displayed at different scales, we take $R = R_s$ and solve for updated $c_i$ or $c_{s,i}$ terms.

The optimization target (recall, on the unit cell scale, identified above as $f(\Delta x) = \Delta x - 87\Delta x^2 + 1778\Delta x^3$) can therefore be expressed on the spring length scale as $f(\Delta x_s) = \Delta x_s - 41.064\Delta x_s^2 + 396.11\Delta x_s^3$, by setting $R = R_s$ and solving for $c_{s,i}$. The final optimized structure, shown in Fig. 5b, c, achieved a simulation force-displacement law on the spring scale of $f(\Delta x) = \Delta x_s - 40.712\Delta x_s^2 + 397.535\Delta x_s^3$, and an experimental one of $f(\Delta x) = \Delta x_s - 40.654\Delta x_s^2 + 397.274\Delta x_s^3$ (shown in Fig. 5d), resulting in an experimental percent difference between targeted and obtained polynomial ratios of 0.294% for the third order ratio and 0.998% for the second order ratio.

## Experimental validation of peak transmitted kinetic energy minimization

In our experimental realization of the chains, the springs with length $a_s = 59$ mm were chosen and incorporated into the rigid frame and connector, resulting in a unit cell length $a = 125$ mm. A chain of twenty unit cells (a single unit cell is shown in Fig. 5c) was fabricated and hung from a frame in order to minimize friction, as shown in Fig. 5e (see Supplementary Note 9 in the Supplementary Information for a more detailed description of the chain design). Impact tests were undertaken with an impactor mass of $M = 40$ g and a velocity of $V = 1.37$ m/s (based on the realized chain, corresponding to the normalized impactor conditions denoted by the dot in Fig. 4a). Data was collected through the use of several cameras positioned along the length of the chain, allowing digital image processing to be used to track the impact wave across the length of the system. In addition, a laser Doppler vibrometer (see Supplementary Information Note 9) was pointed at the last unit cell in the system, allowing for a second measurement of the velocity of the last unit cell. Impact tests were repeated several times. A similar chain of twenty linear unit cells, with similar linear stiffness and mass (the mass of the linear and nonlinear unit cells are 400.4 g and 398.8 g, respectively) values, was then constructed to act as a control for comparison against the nonlinear chain (see Supplementary Information Note 8 for more details), and the impact tests were repeated. It is noted that for the kinetic energy transmission ratio considered herein, the magnitude of the linear stiffness (even if different between the linear and nonlinear chain) does not matter (see Supplementary Information Note 10).

Several key metrics were examined to confirm the performance of the system, the results of which are summarized in Fig. 6. Foremost among these results is the velocity (or kinetic energy) which was transmitted to the end of the chain. Measured spatiotemporal kinetic energy responses are shown in Fig. 6a–d, in which we can see that the nonlinear cases dissipate and trap kinetic energy through unit cell snapping, preventing much of it from reaching the right (or protected) end of the chain (a large portion of the kinetic energy is seen to remain, reflecting back and forth, in the first 9–11 unit cells in panels A and B). We note an excellent match between simulation (using the experimentally fit coefficients taken from the quasi-static force-displacement curve shown in Fig. 5d) and experiment in these spatiotemporal plots. Time histories of measured velocity at the fifth and the last unit cell can be seen in Fig. 6e, f, highlighting the greatly reduced last particle velocity in the nonlinear case as compared to the linear case. Figure 6g shows a comparison of the ratio (of linear to nonlinear cases) of the maximum kinetic energy seen at each unit cell, in both experiment

(averaged across three trials) and simulation. We note that the ratio shown in Fig. 6g is the inverse of $KE$ ratio, for ease of visualization, such that larger numbers denote better impact protection. We see a larger discrepancy at the last two particles (Fig. 6g), which we attribute to non-ideal boundary conditions. We note that damping characterization (see Methods section) suggest the damping in both chains is similar ($\zeta$ of 0.005 for nonlinear and 0.003 for linear). Further, we show in Supplementary Information Note 8, that were the linear chain to have higher damping, e.g., $\zeta = 0.006$, this would have negligible effect on the $KE$ ratio. This highlights the value of nonlinear wave manipulation, where superior performance can be seen without heavy reliance upon damping.

A point of particular note, is that although the targeted conditions show excellent predicted performance, the behavior can be sensitive to small variations in impactor mass and velocity, as seen in the simulation data of Fig. 6h. The variability of the mass-velocity space is immediately apparent, with several very small regions of excellent performance (low $KE$ ratios) surrounded by oscillating regions of lower performance (relatively higher $KE$ ratios), and even several points of poor performance ($KE$ ratio $> 1$). While the region targeted in this work sought a region with relatively low drops in performance (relative to other regions explored via simulation), there still exists varied performance impact conditions nearby. In order to more quantitatively describe this sensitivity, in Supplementary Information Note 11, we calculate the gradient of Fig. 6h, which shows $|\partial(\log_{10}(KE \text{ ratio}))/\partial M|$ can reach near 1 $g^{-1}$ and $|\partial(\log_{10}(KE \text{ ratio}))/\partial V|$ can reach up to 40 s/m. This means that, for the most sensitive regions of the impact conditions landscape, a change in 1 g of impactor mass can result in an up to $\sim 6.7 \times$ change in $KE$ ratio, or a 0.01 m/s change in impactor velocity can result in an up to $\sim 2.5 \times$ change in $KE$ ratio. However, as with the case of the sensitivity to the nonlinear stiffness parameters, near our chosen optimum, the sensitivity is significantly lower, with $\partial(\log_{10}(KE \text{ ratio}))/\partial M = 0.16$ $g^{-1}$ and $\partial(\log_{10}(KE \text{ ratio}))/\partial V = -7.8$ s/m.

This raises the question of why is this nonlinearity so sensitive to impactor conditions. Nonlinear systems in general are known to be sensitive to small changes in system parameters, including initial and boundary conditions, particularly near points of bifurcation or instability[49] (e.g., the butterfly effect[17]). In our case, we suggest this sensitivity is due to two factors: i) The bistability present in the identified optima studied in experiment, and ii) Our choice of performance figure of merit. Considering the effects stemming from bistability, in Supplementary Information Note 11 we show that large, cyclical, changes in performance are seen with with small changes in impactor velocity, where these changes correlate with the arrest of the initially generated solitary wave (as seen in Figs. 1b, 2d, e, and 6a, b) moving one unit cell closer to the boundary opposite the impactor. Further, this solitary wave takes the form of snapping and unsnapping of each unit cell in sequence until, generally (as can be seen in Supplementary Information Note 11, Supplementary Fig. S25), a unit cell remains snapped shut, which corresponds to the point of solitary wave arrest. This is consistent with the idea that with increasing impactor energy, the final unit cell that snaps shut locks in more and more energy until it reaches a critical point, at which it suddenly unsnaps, releasing energy back into the system in the form of kinetic energy. This effect is further amplified, as per our second suggested factor, because we choose peak kinetic energy at the last particle in the chain. We suggest that some of the identified sharp changes in performance are due to the effect seen in Fig. 2d, e, wherein the "best" case of Fig. 2d shows solitary wave arrest just before reaching the boundary, whereas the slightly different nonlinearity of the "bad" case of Fig. 2e allows the solitary wave to interact with the boundary. We expect similar phenomena occur due to the cyclic advance of solitary wave's arrested position with increased impactor velocity. In addition to these factors, we also expect a contribution from wave interference, where small bits of reflected energy can push a unit cell above or below snapping and unsnapping thresholds.

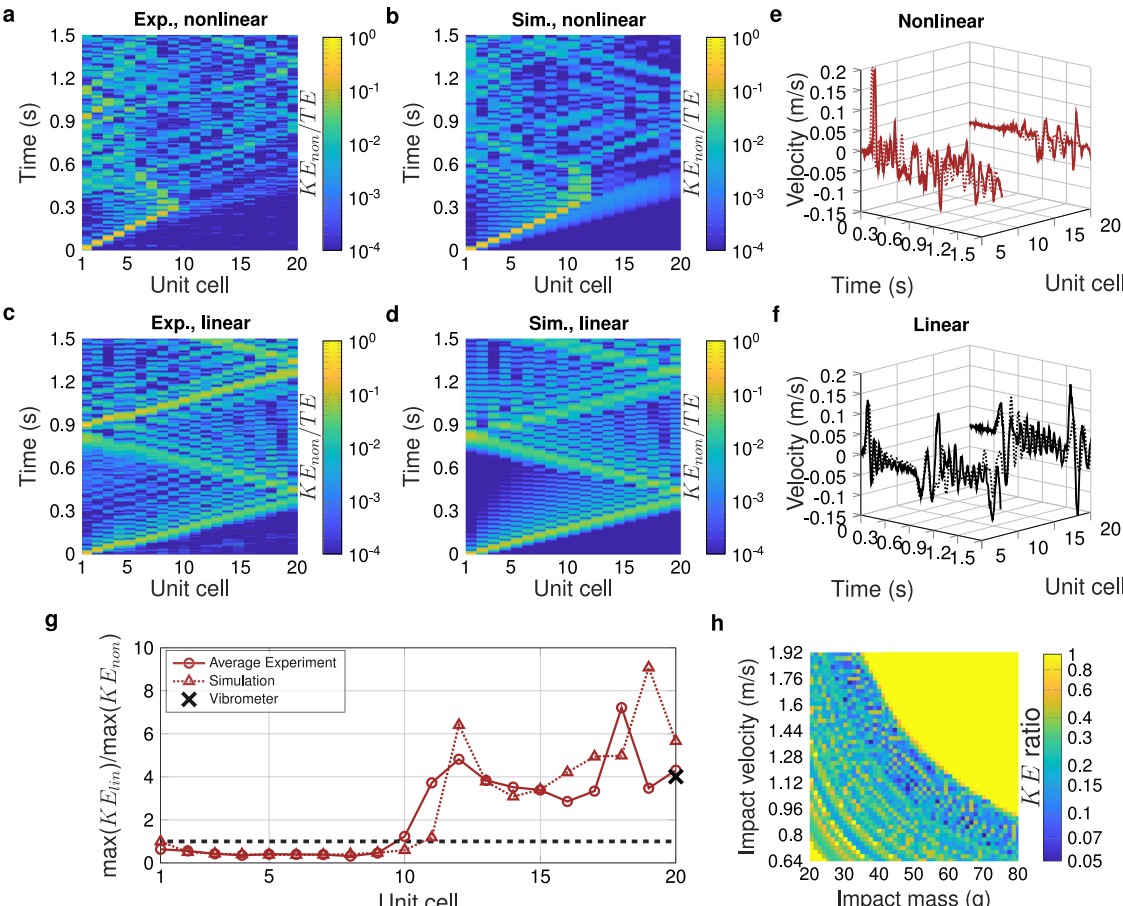

**Fig. 6 | Experimental validation of the optimal nonlinear and linear chains, compared with simulation.** The optimal nonlinear chain corresponds to that identified in Fig. 4a. **a–d** Spatiotemporal evolution of kinetic energy in the system (normalized by the input kinetic energy, or initial total energy). The nonlinear chain is shown in a (experiment, nonlinear trial 1) and b (simulation), while the linear case is shown in c (experiment, linear trial 1) and d (simulation). The experimental spatiotemporal plots include a smoothing of displacement values to assist with noise induced by differentiating the discrete time camera data. The spatiotemporal plots for other trials are included in Supplementary Information Note 9. Time histories of the experimentally measured (solid lines) and simulated (dashed lines) velocities for the fifth and last unit cell in the nonlinear (**e**) and linear (**f**) chains. **g** A

ratio of maximum kinetic energy (linear/nonlinear) seen at each unit cell for both experiment and simulation. A value greater than 1 (denoted by the horizontal dashed black line) indicates superior performance of the nonlinear chain as compared to the linear. The experiment values are the average taken from three experimental trials. The X marks the average experimental value recorded by the vibrometer (see Supplementary Information Note 9 for the full data sets), which collected data from only the last unit cell. **h** The simulated sensitivity of the *KE* ratio (truncated at 1) to impact conditions, wherein the simulation was run with the coefficients that were found experimentally from the nonlinear spring in Fig. 5D (i.e., the physically achieved coefficients). Source data are provided as a Source Data file.

As pertains particularly to the experimental results, we note that the impactor velocities obtained herein were not precise (ranging from 1.36–1.41 m/s, see Supplementary Information Note 9), and this, coupled with the aforementioned sensitivity provides an insight into variations in chain performance—namely, that the slight variations in impactor velocity we see in experiment have the potential to easily knock the system out of its optimal performance region, into one in which poorer performance is to be expected. This phenomenon reveals important characteristics regarding the sensitivity of the system, and more generally nonlinear dynamical systems wherein bifurcation can cause sharp changes in behavior[49]. We believe the proximity of regions of poor performance to regions of good performance (e.g., Fig. 6h) motivates a consideration of nearby conditions in future work. For instance, we expect there are application scenarios in which a region of reduced sensitivity to changes in stimuli may be desirable at the expense of slightly lowered performance. Despite the sensitivity of the system, however, per Fig. 6g the kinetic energy ratio ($KE_{lin}/KE_{non}$) remains greater than 1 (superior performance of the nonlinear chain compared to the linear) for both simulation and experiment once a

critical number of unit cells has passed (unit cell 11 for the simulation, unit cell 10 in experiment).

Before proceeding onto the second case study, we use this peak transmitted kinetic energy minimization problem as an opportunity to remark on the potential multidirectionality of our mulitlevel optimization process. As our method forms a loop (see Fig. 1), in which inputs from the DEM and shape optimization feed back into one another, we note that the design process can be followed in either direction. For example, while one may begin with a desired impact condition, which is then fed into the DEM simulation followed by the mesostructure optimization, one might just as easily begin with some nonlinearity and feed this into the DEM simulation to explore what impact conditions it might perform well (or poorly) for.

## Application of method to a second problem: pulse shape transformation

The bilevel inverse design approach can be applied to different problems of interest by altering the dynamic objective and finding the corresponding optimal nonlinearity. As such, to demonstrate the

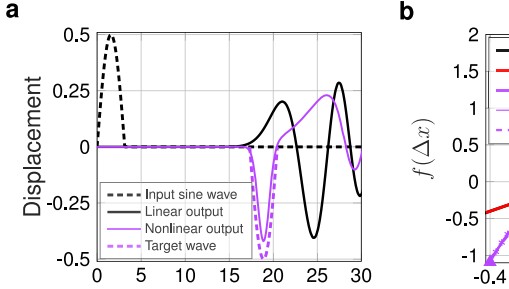
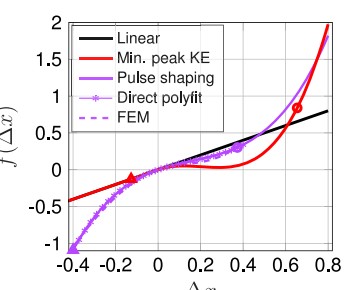
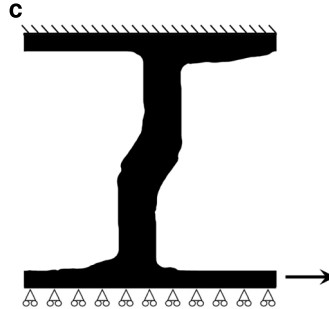

**Fig. 7 | Application of our method to passive signal transformation. a** Input (dashed black) and target output (dashed purple) half-cycle sine waves, and the output waves obtained from DEM simulations with linear (solid black) and optimized (solid purple) nonlinear springs. **b** Force-displacement relations for the pulse shaping (solid purple) shown in (**a**), in contrast with the previous target of KE minimization (solid red), from DEM-optimization. KE minimization resulted in a snap-through mechanism while the pulse shaping displays a softening-to-stiffening response in compression and stiffening response in tension. Direct polyfit obtained from the FEM shape optimization (a line of purple stars) and the validation of the resultant shape via COMSOL FEM (Finite Element Method) simulations (dashed purple). Circle and triangle markers indicate the maximum displacements in compression and tension, respectively. **c** Mesostructure shape identified by the optimizer to match the DEM-identified nonlinearity. Source data are provided as a Source Data file.

potential breadth of applicability of our model, we pursue a second problem: passive signal transformation, wherein we supply an input pulse to the material and seek to transform it into a desired shape at the opposite side. We perturb the system at the left boundary via a prescribed half-cycle sine wave displacement and seek to flip the sign of the pulse at the opposite end of the material, but maintain the same period and amplitude as the input, as shown with dashed lines in Fig. 7a. The optimal nonlinearity, identified using a gradient-based search algorithm (see Methods), which flips the input pulse as seen in Fig. 7a, is found to be $f(\Delta x) = \Delta x - 2.35\Delta x^2 + 4.92\Delta x^3$ and shown in Fig. 7b. The polynomial constitutive law here takes a fully nonlinear form, both in compression and in tension, as opposed to the previous examples with a linear tension regime. The importance of this can be understood in the context that the flip in the sign of displacement before pulse reflection off the opposite end of the system suggests tensile behavior. The qualitative difference between the identified optimal force-displacement curve for the pulse shape transformation and peak transmitted kinetic energy minimization problems can be seen, wherein the pulse shape transformation solution exhibits a relatively slight softening to stiffening behavior in compression and stiffening in tension.

Taking the DEM-identified nonlinearity as our target, we perform shape optimization to identify a unit cell shape to achieve this nonlinearity, this time using a neo-Hookean material model (under plane-strain conditions) to account for the large strains (up to 40%) in both directions, as seen in Fig. 7b. We note that unit cells undergoing geometric nonlinearity that were optimized using a similar process for different quasi-static nonlinearities have been previously realized from silicone confined in rigid plastic frames, and found to be in good agreement with predictions from the neo-Hookean model (see e.g., ref. 24). The resultant optimized geometry is shown in Fig. 7c and the corresponding force-displacement relations obtained via FEM simulations, and validated with COMSOL, in Fig. 7b. We note that such a pulse transformation objective is similar to the formation of rarefaction waves in response to a compressive impact, which has previously been demonstrated in ref. 45, wherein qualitatively analogous softening-in-compression was used. We suggest that this similar result is of particular interest for two reasons: i) Our resulting identified microstructure unit cell geometry is qualitatively different than the origami solution (e.g., our identified solution is in 2D instead of 3D, and not involving creased origami joints), and ii) Our optimization method found a force-displacement response with somewhat qualitatively overlapping features (initial softening in compression) from a random initial guess. We believe the latter point signifies the future potential

for our method to identify previously unknown nonlinear dynamical mechanisms, such as was demonstrated by the optimum nonlinearity identified in the first problem, shown in Fig. 2.

## Discussion

As suggested by the physically large size of our experimentally realized system, we note that this class of geometrically-nonlinear-microstructure-enabled wave transforming materials demands a particularly challenging set of manufacturing requirements. This is due to a particularly large scale separation between the system size and the smallest features. For instance, the optimized design shown in Fig. 5b contains very thin hinges at the top and bottom of the central beam, particularly compared to the size of the unit cell. Given that the wave-dominated behavior shown herein requires many unit cells, we have a situation with three distinct separated length scales (smallest feature, unit cell, system). While this length scale separation is in some sense shared by other metamaterial and lattice structures[50,51], it is amplified for our case. For instance, at the unit cell level, reduction in this disparity is possible, however it comes at the expense of limiting the range of nonlinearity achievable (i.e., one often needs long, thin, deformable structures to achieve large nonlinear response within elastic regimes). Future incorporation of plasticity, phase transformation, or contact may address this issue. Further, recent works have shown the existence of a nonlinear dynamic "size effect" where a critical, minimal number of unit cells has been shown to yield enhanced performance[48], in addition to the longer system size compared to the characteristic wavelengths giving more time and space for traveling pulses to evolve. Indeed, relatively coarse parameter sweeps of the cubic polynomial coefficients for the peak transmitted kinetic energy minimization problem for more highly discretized materials (e.g., 100 unit cells, see Supplementary Information Note 7), suggest the potential for over two orders of magnitude improvement in kinetic energy transmission via the use of a nonlinear material compared to a comparative linear material. All of these length scale separation manufacturing challenges would be further exacerbated in 3D, with the addition of loss of the mechanism for tuning the linear stiffness via out-of-plane structure depth. However, we note that recent advances in 3D printing have seen large jumps in the system to smallest feature size ratio versus manufacturing speed[52]. Self-assembly-based manufacturing is a further tantalizing possibility[53].

While we have demonstrated our design method on a relatively simple (1D, elastic regime, large, few unit cell) system, and extracted several fundamental insights for our primary case study of peak transmitted kinetic energy minimization, we suggest this provides a

basis for future research and technological development. Most directly, within the primary case study of impact mitigation, a near term question of future interest would be how such optimal nonlinearities would change for a different metric such as maximum transmitted force or peak tensile stress anywhere in the material. Similarly immediate, given the flexibility of a designer nonlinear spring and the incorporation into a lumped-mass context shown herein, we suggest this enables the possibility for rapid experimental investigation of nonlinear dynamical phenomena only seen thus far in simulation and theory. Less directly, one can imagine extending the bilevel optimization to 3D and to more complex constituent material models. The extension to 3D yields particularly non-trivial challenges, due to the second-order tensorial nature of solid mechanical systems, including with the potential for shear-to-longitudinal mode conversion in linear[54], and not-to-mention-nonlinear[55], dynamical regimes. With regards to optimization of unit cells in 3D for desired quasi-static nonlinear mechanical response, some initial steps have recently been made for select multiaxial loading states[56]. Extension of this method to less scale separated dynamical (no lumped masses) would enable more mass efficient structures, but would likely required either some degree of homogenization or larger, high-performance computing capability. The incorporation of irreversibility of the constituent material model, such as plasticity, would open enhanced energy dissipation mechanisms and application to high strain-rate applications[57]. Similarly, the incorporation of activity or stimuli-responsivity[58] in the constituent material model would open applications in shape morphing, mechanical computing, and the capacity for materials that autonomously respond and conduct directed work[51]. The use of shape memory materials[58] offers potential in each of these capacities, via targeted shape change, potential resetting capability, and enhanced energy absorption. Incorporation of multiphysics coupling into the material model may create additional possibilities in fields such as optomechanics[59].

In each of the above cases, in addition to opening new application possibilities, we believe that this method has the potential to help answer fundamental questions concerning the interplay of nonlinearity with these varied mechanisms, and most generally the spatiotemporal partition of energy in nonlinear dynamical systems[38]. Such nonlinear dynamical questions increase in complexity greatly, when considering the influence of disorder[60], including in the form of designed defect placement or spatially extended heterogeneity. It also raises fundamental questions, and via this method a possible new avenue for answering, in the form of design of nonlinear systems in the context of stimuli sensitivity. For instance, in our case of peak transmitted kinetic energy minimization, we observed (as common for many nonlinear systems) large sensitivity of the kinetic energy transmission metric to small changes in nonlinearity and impact conditions. This indicates future possibilities where computational design can incorporate stimuli sensitivity and parameter uncertainty into the design objective. Further, one can also imagine the potential to leverage the tools developed in the nonlinear dynamical systems community. Instead of optimizing for features in the time domain response of the system, one may optimize in the context of bifurcation diagrams[61] or phase space[49], where in the case of the former, new modes emerge and disappear with variation of input parameters or change stability, and in the latter, nonlinear dynamical objects such as limit cycles and strange attractors are formed. Noting the capacity for systems such as ours to be described via partial differential equations and continuum approximations (most directly, such as the KdV description of the FPUT system[35]), analysis tools developed for these limits[17] may be incorporated into optimization objectives to improve the ability for the computer to meaningfully traverse and efficiently search the design space. Considering all these possibilities together, results in a rich array of possible future extensions.

## Methods

### DEM simulations
We numerically integrate nondimensional Supplementary Eq. (S10) in Supplementary Information Note 3 using a Runge-Kutta algorithm (ode45 in MATLAB) with impactor velocity $V/V_0$ and mass $M/m$ applied to the impactor particle. The output and maximum integration time-step is selected by estimating the highest nondimensional frequency $f_{max} = \frac{1}{2\pi}\sqrt{\frac{k_{max}}{M_0}}$, where $k_{max} = \max(1, 1 + 2c_2 + 3c_3)$, such that the time-step $\Delta t = 1/(1000 f_{max})$. The nondimensional displacement and velocity tolerances are set to $10^{-10}$. The total energy conservation is checked for an undamped DEM system, and deviation is less than 0.1% over the entire simulation duration.

### Optimization of nonlinear constitutive law
We search for the optimal nonlinear spring coefficients $(c_2, c_3)$ in Eq. (2) to minimize a dynamic performance metric, which results in a constrained nonlinear optimization problem in the form

$$\min_{c_2, c_3} J(\boldsymbol{x}(c_2, c_3)) \text{ subject to } W(c_2, c_3) \geq 0, \qquad (4)$$

where $J$ is the objective function which depends on the DEM simulation trajectory $\boldsymbol{x}$, and $W$ corresponds to the strain energy constraints explained in the Supplementary Information Note 4. For most of this text, the objective is chosen as the normalized peak kinetic energy at the end of the material, $J = \max(KE_{non})/\max(KE_{lin})$, nevertheless the methodology can be applied to other passive wave transformation applications. In the pulse shape transformation example, Fig. 7, the objective is chosen as the $L^2$-norm of the error between the target and the output pulse within the half-period of the input wave, i.e., $J = ||\boldsymbol{x}_{tgt}(\boldsymbol{t}) - \boldsymbol{x}(\boldsymbol{t})_{out}||_2$ where $|\boldsymbol{t}| = T_{inp}/2$. The design space is explored using a gradient-based optimizer, fmincon in MATLAB, where the gradients are estimated using finite differences and the Karush-Kuhn-Tucker (KKT) conditions are used to ensure optimality of the solution.

### Structural optimization
We include here further details about the shape optimization of the spring geometry, while noting that a full description of the method can be found in our previous work[24]. A level set method is used to perform a nonlinear, displacement control, 2D continuum shape optimization.

For the case of minimizing peak transmitted kinetic energy, the spring is modeled using a 2D plane stress condition, with a Kirchhoff material model (linear elastic material model, with geometric nonlinearity included), as we have found this material model to be a good representation of polycarbonate throughout displacement ranges which do not reach the plasticity threshold of the material. The optimization performed herein was solved with a uniform mesh consisting of 750 × 750 quadrilateral elements. A symmetry boundary condition (rollers) is applied along the bottom edge of the optimization domain, such that only one half of the spring need be simulated. The top edge of the domain has a fixed boundary condition enforced. A 1D applied displacement is applied at the bottom right edge of the domain, where the connector to the adjacent spring would exist in the physical system. These conditions, along with the initial condition supplied to the optimizer, are illustrated in Supplementary Note 8 in the Supplementary Information.

Prior to manufacture and experimental tests, the force displacement output from the optimization process was further confirmed via a FEM simulation in COMSOL Multiphysics. Following the verification process outlined in our previous work[24], the optimal level set was converted to a 2D geometry in COMSOL, and a quasi-static simulation was performed. This simulation similarly used 2D plane stress, with a Kirchhoff material model and the same applied boundary conditions

as the optimization process. The COSMOL simulation, however, employed a body-fitted mesh consisting of 20,447 triangular elements. The results, shown in Fig. 5D, confirm the accuracy of the optimization simulation.

As outlined in Eq. (3), the objective function for the optimization is taken to be the ratio of polynomial coefficients resulting from the force-displacement curve of the current design. This polynomial is fit to the force-displacement curve using the equation $c = V^{-1}C$, in which $V$ is the Vandermonde matrix, and $C$ is compliance (we note that herein, we use lowercase $c$ to refer to polynomial coefficients, and uppercase $C$ to refer to compliance). Fitting to compliance, rather than force, is done for ease of sensitivity calculation, as compliance optimization is a well known optimization problem. Noting that $C = F\Delta x$ (in which $\Delta x$ is displacement), it is a simple matter to transition between force and compliance. We can then write the sensitivity for the polynomial ratio objective as a combination of known compliance sensitivity terms and sensitivity terms related to the fitting of the polynomial.

$$\frac{\partial c_i}{\partial \Omega} = \sum_{j=1}^{m} \frac{\partial c_i}{\partial C_j}\frac{\partial C_j}{\partial \Omega} = \sum_{j=1}^{m} V^{-1}(i,j)\frac{\partial C_j}{\partial \Omega}, \qquad (5)$$

where $\Omega$ refers to the level set domain, $i$ refers to the degree of polynomial, $j$ refers to the displacement step, and $m$ is the total number of displacement points under consideration (see our previous work for more details[24]).

We note that, as our polynomial fit represents an element in equilibrium at rest, the zeroth order term must be taken to be zero in order to have a physically consistent and realistic polynomial fit. The equation $c = V^{-1}C$ may sometimes result in small but nonzero zeroth order terms, leading to significant differences between the fit coefficients and the actual system behavior. We therefore force the zeroth order term to be zero by assuming $c_0 = 0$, which effectively zeros out the last column of V. Similarly, due to the number of displacement points being higher than the order of polynomial being fit, higher order polynomial terms (higher than three, in this case) can be assumed to be zero, thus effectively zeroing out the corresponding columns in $V$. The $V$ matrix is therefore frequently non-square (as is the case for the bistable optimization example shown herein). We therefore use the Moore-Penrose inverse (or pseudoinverse)[62] to calculate $V^{-1}$.

Shape optimization for the pulse shape transformation problem is carried out with ParaLeSTO[63], an open source level set topology optimization code, where FEniCSx[64–67] is used as the FEM platform with SNES solvers from PETSc[68]. The material experiences especially large strains, up to 40% in both tension and compression as seen in Fig. 7, therefore we chose the neo-Hookean hyperelastic material to model the elastic response. Nonlinear, quasistatic, displacement controlled FEM simulations are performed under 2D plane strain assumptions by using a uniform, 100 × 100 quadrilateral mesh. Note that due to the weaker nonlinearity of this case, compared with the highly nonlinear bistable spring, a coarser mesh was able to accurately capture the mechanical response. Compliance sensitivities $\partial C_j/\partial \Omega$ are computed at each displacement step $j$ using the discrete adjoint method, which are then modified by the polynomial fitting terms of Eq. (5) to obtain the final sensitivity.

## Unit cell design and manufacturing
After the nonlinear spring had been designed via optimization, the design was resized to the experiment scale (wherein the length of one spring, $a_s$, was taken to be 59 mm). The single spring design was reflected over the symmetry condition, and in order to minimize the non-longitudinal motion of the masses, a second set of nonlinear springs was added in parallel to the first. A single unit cell (shown in Fig. 5c) was fabricated by cutting the shape out of a polycarbonate sheet via computer-numerical-control (CNC) milling. This milled

shape was then sanded down at the edges to fit smoothly into a rectangular stainless steel frame, which was commercially manufactured via sheet cutting, and secured in-plane with an acrylic backing bolted into the frame and glued to the milled spring (see Supplementary Note 9 in the Supplementary Information for a visualization). We note the design of the frame surrounding a spring is particularly important in properly imposing boundary conditions, and thus critical in matching simulation predicted force displacement curves. The choice of a stainless steel frame herein was made in order to meet these criteria, while still allowing for the milling of the spring to fit within the manufacturing constraints of the manufacturing equipment.

A single unit cell was tested quasi-statically to confirm the performance of the spring, as shown in Fig. 5d. Repeated, cyclic quasi-static loading tests were also performed to confirm that no onset of plasticity or fatigue occurs during the dynamic experiment (see Supplementary Information Note 8).

## Data collection
The motion of each unit cell was captured with the use of cameras mounted above the chain. The length of the chain necessitated the use of multiple cameras to capture the motion all unit cells. Each experimental trial therefore has four videos associated with it, shot in slow motion at 240 fps on four iPhones, with each video overlapping by at least one unit cell to ensure proper spatial synchronization and the capture of all motion. For each trial, the cameras and unit cells were spaced such that camera 1 captured unit cells 1–6, camera 2 captured unit cells 4–11, camera 3 captured unit cells 9–16, and camera 4 captured unit cells 14–20. Video processing was performed in MATLAB. Temporal synchronization between each video was achieved by a visual signal which caused a fluctuation in intensity of light, which could then be detected in each video and used to synchronize the times between the recording devices. Tracking of each particle was achieved with the use of colored markers on each unit cell. Post processing in MATLAB allowed each frame to be separated into discrete color channels, allowing for the isolation and tracking of red, blue, and green markers (blue was used for the impactor, green for the back edge of each unit cell, and an additional red marker for unit cells which were covered by overlapping camera fields of view, in order to enable spatial synchronization of the data from each camera).

## Damping characterization
In order to characterize the damping ratio $\zeta$ of the manufactured springs, the power spectrum was experimentally measured, generated from a low-amplitude 1 ms duration square pulse excitation applied to a single spring via electrodynamic shaker. A Lorentzian function was numerically fit to the response for both the nonlinear and linear spring cases. The experimental setup for both cases are shown in Supplementary Information Note 8, where the single unit cell was hung horizontally using fishing lines, and the end of the connector, which was connected to the center of the spring via bolts and nuts, was clamped to impose a fixed boundary condition (FBC). The pulse excitation was provided by a function generator (FG, Tektronix AFG3022C), controlled through MATLAB, and applied via an electrodynamic shaker (The Modal Shop K2007E01) equipped with a stinger, whose tip was manually set at a small distance (within 1 mm) away from the side surface of the unit cell. A laser Doppler vibrometer (LDV, Polytec PSV 400) was used to record the dynamical response of the unit cell in velocities. Data was collected by measuring the side surface of the unit cell (averaged three times for each test), where the LDV and the FG were synchronized through a common trigger signal for repeatability of the averaging process. Measured dynamical responses, which were processed to account for the tilted angle between the LDV scanning head and the unit cell, are shown in Supplementary Information Note 8. The normalized power spectrum was obtained by taking the

square of the Fourier transform (FFT) of time domain data and then normalizing its maximum value. The FFT was taken using the built-in *fft* command in MATLAB. The normalized $|FFT|^2$ result was fit to a Lorentzian function of the following form

$$L(f) = \frac{A}{\pi} \frac{\Gamma/2}{(f - f_0)^2 + (\Gamma/2)^2}, \tag{6}$$

where $A$ is the amplitude parameter, $f_0$ is the central frequency, and $\Gamma$ is the full width at half maximum in frequency. The fitting process was done by using the built-in *fit* command in MATLAB. The $Q$ factor was found by taking $f_0/\Gamma$ from the numerical fitting, which are approximately 92 for the nonlinear and 177 for the linear spring unit cell. Based on these $Q$ factors, by using the equation $\zeta = 1/(2Q)$, we calculated the damping ratios $\zeta$ as 0.005 and 0.003 for nonlinear and linear chains, respectively, which were used in the numerical simulations.

## Data availability

Source data are provided with this paper. Two sets of four videos (reduced resolution and sped up via downsampling by 8 × to meet file size limitations) corresponding to the data shown in Fig. 6a, c are included as part of the Supplementary Information. The same naming convention is used as in Supplementary Information Table 1, but with the suffix "_SpedUpLowRes". All related videos regarding the data in the main text are uploaded to: Boechler, Nicholas (2024), "Customizable wave tailoring materials enabled by nonlinear bilevel inverse design 1", Mendeley Data, V1, https://doi.org/10.17632/2wgwfy2wfg.1. Videos related to the data in Supplementary Information Note 9 are uploaded to: Boechler, Nicholas (2024), "Customizable wave tailoring materials enabled by nonlinear bilevel inverse design 2", Mendeley Data, V2, https://doi.org/10.17632/6bg6hr5kyr.2. SOLIDWORKS and level set files of the optimized spring design shown in Fig. 5 are uploaded to Boechler, Nicholas (2025), "Customizable wave tailoring materials enabled by nonlinear bilevel inverse design 3", Mendeley Data, V1, https://doi.org/10.17632/fspk9kw98v.1. Source data are provided with this paper.

## Code availability

Sample DEM MATLAB script (DEM_example.m) is uploaded with the Supplementary Information. Topology optimization codes that generate the results will be made available upon request. The level set topology optimization code for the impact mitigation problem was based on OpenLSTO, which is available at https://m2do.ucsd.edu/software. The level set topology optimization code for the pulse shape transformation problem was based on ParaLeSTO, which is available on GitLab at https://gitlab.com/m2dO1/paralesto.

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

## Acknowledgements

This work was supported by the UC National Laboratory Fees Research Program of the University of California, Grant Number L22CR4520. B.M. acknowledges support from the U.S. Department of Energy (DOE) National Nuclear Security Administration (NNSA) Laboratory Graduate Residency Fellowship (LRGF) under Cooperative Agreement DE-NA0003960. I.F. acknowledges support from the Department of Defense (DoD) through the National Defense Science & Engineering Graduate (NDSEG) Fellowship Program. M.B. acknowledges support from the National Science Foundation Graduate Research Fellowship Program under Grant No. DGE-2038238.

## Author contributions

N.B. and H.A.K. designed research. B.M., H.X. I.F., K.Q., C.T., M.B., N.B., and H.A.K. performed research. K.Q., B.M., and I.F. conducted the experiments, and B.M., H.X. I.F., K.Q. analyzed the results. B.M., H.X. I.F., K.Q., C.T., M.B., N.B., and H.A.K. wrote the paper.

## Competing interests

The authors declare no competing interests.
