## [Transparent Peer Review file · Nature Communications]

Customizable wave tailoring nonlinear materials enabled by bilevel inverse design

Corresponding Author: Professor Nicholas Boechler

Version 0:

Reviewer comments:

Reviewer #1

(Remarks to the Author)

The paper describes a computational and experimental investigation of a nonlinear one-dimensional chain of oscillators, representing a prototype metamaterial aimed at vibration control. The nonlinear behavior of the oscillators is designed to minimize the kinetic energy transmitted along the chain and generated by an impact applied at one end. This nonlinearity is characterized by a polynomial force-displacement relationship with cubic and quadratic terms, and a nonsym-metric response in tension-compression. The mechanical system is numerically investigated using direct integration of the first-order ODEs. An optimization of the constitutive polynomial relation-ship is performed to localize the kinetic energy, thereby arresting wave propagation along the chain. The numerical results are validated with an experimental campaign, where the chain is re-produced with nonlinear oscillators whose constitutive behavior was obtained through shape op-timization exploiting geometric nonlinearities.

The investigated topic is very interesting and relevant to current research on metamaterials for vi-bration mitigation. Several research groups are working on this topic, with the main goal of proper-ly exploiting nonlinearities to extend linear theory and improve performance, paving the way for real applications. The experimental campaign illustrated in the paper is valuable, particularly con-sidering the manufacturing of the nonlinear oscillator and the realized setup. However, it is the reviewer's opinion that the paper is not suitable for publication in the journal for the reasons de-tailed below.

Literature Review

The literature review presented in the paper is well-framed according to the different tasks per-formed: nonlinear wave propagation in various physical contexts, and shape optimization for achieving a defined force-displacement nonlinear behavior. However, a literature review focused on the main target of the investigation, i.e., tuning the nonlinearity to create a bandgap in the propagation of mechanical waves, is missing. A reader expects to see a careful review of the state of the art, highlighting what has been done by other research groups and the new findings achieved with the investigation. The literature review is too general and does not address the main claim of the work.

Results

The presented findings do not exhibit scientific generality and appear to be a parametric investiga-tion of the polynomial coefficients of the nonlinear oscillators. Consequently, the results hold technical but not scientific value. For instance, why is a nonlinear polynomial force-displacement relationship assumed? Why does this relationship include quadratic and cubic terms? Why is a non-symmetric behavior adopted in tension and compression? These aspects should be addressed and justified from a mechanical point of view. Moreover, while a partial interpretation of these choices is provided from a static perspective, an analysis of the nonlinear interaction and explana-tion of the associated phenomena should be provided. The results presented in Fig. 2 are valuable and demonstrate the efficacy of the performed opti-mization. A description of the role of the parameters is provided with reference to the static force-displacement relationship. Additionally, the sensitivity of the response as a function of the param-eter is analyzed, but a description of the dynamical behavior is missing. What happens to the kinet-ic energy from cells 1 to 15? What mechanism stops the propagation of the mechanical waves? Is there a localization phenomenon? Is the energy transferred back to the source? A proper dynam-ical analysis of these aspects, supported by results (e.g., time response of the oscillator, analysis of the frequency content, etc.), is missing. Can the optimization create a bandgap for wave propaga-tion? Is it possible to extract a dispersion curve that clarifies the wave-stopping phenomenon? These questions should be clearly addressed considering the topic and the claims made.

Experimental Campaign

The performed activities have great experimental value. However, a more accurate analysis of the observed response is missing. Considering the advanced experimental setup employed, it would have been useful to show the time history of the response to clarify the dynamical interactions. A discussion about the practical limitations of realizing this kind of nonlinear chain in a new material is important, considering that this seems to be the main problem to overcome. In conclusion, it is the reviewer's opinion that the paper cannot be published in its current form because the main claim is not properly addressed. The content is interesting and relevant but is more focused on the different tasks performed. A broader perspective, from which general and structured findings emerge, is missing.

Reviewer #2

(Remarks to the Author)

In this article, the authors describe a way in which the nonlinearity of a spring-mass system can be optimized to minimise the kinetic energy at one end of the spring-mass system given an initial 'jolt' to the system at the other end. This is achieved by the solution of an discrete model using numerical methods that is used to inform the design of a physical material in an experiment that also minimises the kinetic energy.

Firstly, I'd like to congratulate the authors on an really well-written article, I didn't spot a single typo, it was written clearly, and, I believe after an appropriate amount of study on my behalf, to have understood what they are trying to achieve here. I'd like to emphasise that I see nothing wrong with their findings, they were extremely transparent with their methodology and I have no reason to doubt the validity of their work. I also found it really interesting, so well-done.

However, I do have some reservations with regards to whether this article may be better suited to a different journal, as I do not believe the authors have convinced me this is important enough for inclusion into Nature Comm, I think that perhaps Sci. Rep. may be a more appropriate place for it. Whilst I enjoyed the article and was impressed by the results, I was left with a feeling of 'so-what?' which I think the authors could've done a better job of setting the future context.

I have the following comments:

- 1) Equation (1) is an adaptation of the famous Fermi-Pasta-Ulam-Tsingou problem; I wonder if the authors missed a trick here by not mentioning/referencing such a famous problem (apologies if I missed it). Putting a fresh spin on such an important work in the history of nonlinear mathematics/physics would put this work in a better historical perspective. To that end, I wonder if equation (1) can be reduced to a KdV-type PDE when applying a continuum hypothesis; this would be valid for a large number of cells with a small gap-width and it may be an interesting future direction; could the authors comment on this? Have they considered the FPUT problem and how it links into their results?
- 2) I got a little confused on what the system parameters that have to chosen are and what the solution measures are; it would help the reader, if around line 80, the authors simply stated what the control parameters are and what they will measure.
- 3) By the way, why only a cubic polynomial? I'm not sure this was sufficiently motivated, perhaps the authors could expand on this a little more? I mean presumably the order has to be odd to get bi-stability, and a cubic is the simplest way to achieve this, but perhaps the authors could expand on this a bit more.
- 4) Reading through the article, I think I needed a bit more of an explanation as to how the geometry could be changed of the unit cell to mimic the nonlinearity. I am a mathematician and often struggle with experimental figures (which is my fault I know) but I would've loved a simpler explanation and accompyaning figure to understand how the physical system can be adapated; this is Nature Comm. and should be as accessible as possible. I know there was an explanation in the supplementary material, but, in my humble opinion, I think the authors missed a trick here by not being super-transparent about this in the main text.
- 5) The discussion: This is where I think the paper needed the most work; the future work was described as simply changing the objective function in the optimization problem which I think is rather putting this research down a bit; surely there must be some more important applications which I don't think the authors spelt out clearly enough.

Reviewer #3

(Remarks to the Author)

The work shows an experimental realization of a material with tailored nonlinear constitutive responses. The system is obtained via a shape optimization inverse design and validated experimentally with good agreement. The methodology shown in the work allows for on-demand quasi-static nonlinearity which is of relevance for the community of mechanics. The article is well written, covers a broad audience and reaches the standards of the journal, however there are several points to be revised and improved. I suggest publishing the article once these points will be solved.

1. The general expression of Δx is provided in the Note 3 of the supplementary. I think this expression is important enough to mention it in the main article. In fact, the optimization is based on this. Moreover, the Physical meaning of the

terms in Eq. (1) may be explained for the clarity of the reading. In a general way, I think that the discussion of Note 3 is important enough to be synthesized in the main text. Figs. 2A, 2B and 2C are very difficult to follow without this supplementary material. In my opinion the main text is not enough to understand the results. I encourage the authors to introduce a summary of the Note 3 in the main article.

2. In Eq. (2) Ω is not defined. It is defined later but not here.

3. Dashed line in Fig. 5E is not defined.

4. A short discussion about the sensibility of the shape optimization to the properties of the impactor. Apparently, there is a huge sensitivity to small variations of the impactor.

Version 1:

Reviewer comments:

Reviewer #1

(Remarks to the Author)

The authors have addressed all comments provided by the reviewer. The paper has been improved according to the provided suggestions. The opinion of the reviewer is that this work represents a high-quality computational and numerical study, demonstrating a rigorous and well-executed approach. The translation of known mechanical concepts into an application case is carried out with precision, making this a valuable contribution to the field. While the content does not introduce a groundbreaking discovery, it undoubtedly constitutes an excellent piece of research worthy of publication in a prestigious journal. However, the reviewer expresses some doubts about whether Nature Communications is the most suitable venue for this work, as its focus may be more aligned with journals emphasizing methodological advancements rather than fundamental scientific breakthroughs. Nevertheless, this decision ultimately rests with the editor.

This personal opinion has remained unchanged even after the thorough revision of the article and is mainly based on the following aspects: (i) I do not see a new discovery in the obtained and pre-sented results, especially in the way they are conveyed; (ii) the answers provided to my questions and, more generally, the explanations of the reproduced phenomena rely more on mathematical modeling than on the mechanical understanding of the observed phenomena, which in this context represents the main approach to generalizing the results and making them universal.

Reviewer #2

(Remarks to the Author)

Thank you for considering my comments and providing a detailed revision in response to them. I believe my comments have been answered sufficiently and have no further criticisms.

Reviewer #3

(Remarks to the Author)

The authors have addressed my comments and suggestions. The article has been really improved and reaches the standards of the journal. Most importantly, it is of relevance for a broad community covering among others, nonlinear Physics, metamaterials and mechanics.

Responses to Reviewers (Reviewer comments in bold, our response in blue):

We greatly appreciate the constructive feedback provided by the Reviewers, which has allowed us to greatly improve our manuscript. Below, we provide a point-by-point response to each of their comments along with a synopsis of the revisions that we have applied to the manuscript. A copy of our revised manuscript (the revisions made are also highlighted in blue text in the manuscript) is also attached.

Responses to Reviewer 1:

The paper describes a computational and experimental investigation of a nonlinear one-dimensional chain of oscillators, representing a prototype metamaterial aimed at vibration control. The nonlinear behavior of the oscillators is designed to minimize the kinetic energy transmitted along the chain and generated by an impact applied at one end. This nonlinearity is characterized by a polynomial force-displacement relationship with cubic and quadratic terms, and a nonsymmetric response in tension-compression. The mechanical system is numerically investigated using direct integration of the first-order ODEs. An optimization of the constitutive polynomial relationship is performed to localize the kinetic energy, thereby arresting wave propagation along the chain. The numerical results are validated with an experimental campaign, where the chain is re-produced with nonlinear oscillators whose constitutive behavior was obtained through shape optimization exploiting geometric nonlinearities. The investigated topic is very interesting and relevant to current research on metamaterials for vibration mitigation. Several research groups are working on this topic, with the main goal of properly exploiting nonlinearities to extend linear theory and improve performance, paving the way for real applications. The experimental campaign illustrated in the paper is valuable, particularly considering the manufacturing of the nonlinear oscillator and the realized setup.

We appreciate the careful consideration of our manuscript by the Reviewer, and their positive comments regarding our work.

However, it is the reviewer's opinion that the paper is not suitable for publication in the journal for the reasons detailed below.

In the following, we describe our extensive revisions in response to the Reviewer's comments, which we hope have convinced them regarding the suitability of our work for publication in Nature Communications.

Literature Review

The literature review presented in the paper is well-framed according to the different tasks per-formed: nonlinear wave propagation in various physical contexts, and shape optimization for achieving a defined force-displacement nonlinear behavior. However, a literature review focused on the main target of the investigation, i.e., tuning the nonlinearity to create a bandgap in the propagation of mechanical waves, is missing. A reader expects to see a careful review of the state of the art, highlighting what has been done by other research groups and the new findings achieved with the investigation. The literature review is too general and does not address the main claim of the work.

We thank the Reviewer for pointing this issue out, and have added the following text and references, which we hope will clarify this issue for the reader.

1. In the first paragraph, we added the following review: *Deng, B., Raney, J. R., Bertoldi, K. and Tournat, V. Nonlinear waves in flexible mechanical metamaterials. J. Appl. Phys. 472 130 (2021).*
2. We added a new paragraph in the introduction, as follows, focused on the impact mitigation via nonlinear metamaterials objective with additional literature review focused on this topic. Newly added references are in italicized text.

“Focusing on the problem of minimizing peak kinetic energy transmitted via waves in response to impact, we first note that because impact is inherently a broadband excitation, prior linear mesostructured material (or “metamaterial”) strategies that leverage bandgaps [*Hussein, M. I., Leamy, M. J. and Ruzzene, M. Dynamics of phononic materials and structures: Historical origins, recent progress, and future outlook. Appl. Mech. Rev. 66, 040802 (2014)*] have been shown to have limited efficacy (*e.g.*, requiring gradient or disordered material strategies that increase bandwidth at the cost of attenuation [*Tan, K. T., Huang, H. and Sun, C. Blast-wave impact mitigation using negative effective mass density concept of elastic metamaterials. Int. J. Impact Eng. 64, 20–29 (2014)*]). In favor of this, several nonlinear mesostructured material motifs have been realized and studied in the context of impact, leveraging nonlinearities such as the aforementioned contact (tensionless and stiffening in compression [*Nesterenko, V. Dynamics of heterogeneous materials (Springer Science and Business Media, 2013)*]), tensegrity (stiffening in tension and tunably softening or stiffening in compression [*Fraternali, F., Carpentieri, G., Amendola, A., Skelton, R. E. and Nesterenko, V. F. Multiscale tunability of solitary wave dynamics in tensegrity metamaterials. Appl. Phys. Lett. 105, 201903 (2014)*]), and compressed beams (bistability and snap-through in compression [*Shan, S. et al. Multistable architected materials for trapping elastic strain energy. Adv. Mater. 27, 4296–4301 (2015)*]), as well as o-rings (tensionless with double-power law response in compression [*Herbold, E. and Nesterenko, V. Solitary and shock waves in discrete strongly nonlinear double power-law materials. Appl. Phys. Lett. 90 (2007)*]) and origami (softening in compression and stiffening in tension [*Yasuda, H. et al. Origami-based impact mitigation via rarefaction solitary wave creation. Sci. Advances 5, eaau2835 (2019)*]). Herein, using our method, we obtain the following two new fundamental insights within this area. The first, is that very small differences in the nonlinear spring response (without qualitative difference) can drastically change the response of the system (changing the nonlinear coefficients by less than 14% results in an over 3000% change in performance) in such dynamical settings, from severely under- to outperforming a comparative linear system. To our understanding, this degree of sensitivity was hitherto unknown in the context of impact mitigation. The second, is that the often used design strategy of impact mitigation via “energy locking” bistability [*Shan, S. et al. Multistable architected materials for trapping elastic strain energy. Adv. Mater. 27, 4296–4301 (2015)* and *Fancher, R. et al. Dependence of the kinetic energy absorption capacity of bistable mechanical metamaterials on impactor mass and velocity. Extrem. Mech. Lett. 102044 (2023)*] can be greatly outperformed by our identified optimal snap-through nonlinearity (by over a factor of three times). We then choose a second high performing identified nonlinear constitutive response (which is more amenable to our experimental capabilities) and demonstrate the full inverse design of a superior impact mitigation system, from identification of an ideal nonlinearity, to the design of the unit cell geometry, and experimental validation of the performance. While we focus the majority of the manuscript on this impact mitigation problem, as demonstrated by our application of this method to the aforementioned two qualitatively different objectives, we believe our method will have wide future applicability, including to the previously referenced array of nonlinear acoustic, phononic, and mechanical wave transformation applications [...]

Results

The presented findings do not exhibit scientific generality and appear to be a parametric investigation of the polynomial coefficients of the nonlinear oscillators. Consequently, the results hold technical but not scientific value.

We have attempted to clarify the scientific generality and value via our revisions to the manuscript, which include the following revisions. Foremost among these are:

- 1) The new bilevel design method. To show the generality of this, we have also applied our method to a second, qualitatively different, problem: pulse shape transformation.
- 2) Two fundamental insights, described in the following, concerning impact mitigation with nonlinear metamaterials.

Abstract: “Passive wave transformation via nonlinearity is ubiquitous in settings from acoustics to optics and electromagnetics. It is well known that different nonlinearities yield different effects on propagating signals, which raises the question of “what precise nonlinearity is the best for a given wave tailoring application?” Considering a one-dimensional spring-mass chain connected by polynomial springs (a variant of the celebrated Fermi-Pasta-Ulam-Tsingou system), we introduce a bilevel inverse design method which couples the shape optimization of structures for tailored constitutive responses with reduced-order nonlinear dynamical inverse design. We apply it to two qualitatively distinct problems—minimization of peak transmitted kinetic energy, and pulse shape transformation—demonstrating our method’s breadth of applicability. Focusing on the first problem, we obtain two fundamental insights: i) Small differences in the nonlinearity can drastically change the dynamic response of the system, from severely under- to outperforming a comparative linear system; and ii) The often used design strategy of impact mitigation via “energy locking” bistability can be significantly outperformed by our identified optimal nonlinearity. We validate this case with impact experiments and find excellent agreement. This work sets the foundation for broader passive nonlinear mechanical wave tailoring material design, with applications to computing, signal processing, shock mitigation, and autonomous materials.”

Introduction: “In addition to the introduction of this method, we apply it to two qualitatively distinct problems—minimization of peak transmitted kinetic energy in response to an impact, and pulse shape transformation (inversion of an applied boundary displacement signal at the other end of the chain)—demonstrating the potential breadth of applicability of our method. We highlight also that in both cases, we conduct a comparison between the linear and nonlinear response. We assert that this comparison is particularly important from a fundamental perspective, as it isolates the role of nonlinearity from other linear wave manipulating effects such as dispersion, dissipation, and heterogeneity.” ... followed by the added paragraph referenced in the response to the previous question.

New section—Application of method to a second problem: Pulse shape transformation.

“The bilevel inverse design approach can be applied to different problems of interest by altering the dynamic objective and finding the corresponding optimal nonlinearity. As such, to demonstrate the breadth of applicability of our model, we pursue a second problem: passive signal transformation, wherein we supply an input pulse to the material and seek to transform it into a desired shape at the opposite side. We perturb the system at the left boundary via a prescribed half-cycle sine wave displacement and seek to flip the sign of the pulse at the opposite end of the material, but maintain the same period and amplitude as the input, as shown with dashed lines in Fig. R1A. The optimal nonlinearity, identified using a gradient-based search algorithm (see Methods), which flips the input pulse as seen in Fig. R1A, is found to be $f(\Delta x) = \Delta x - 2.35\Delta x^2 + 4.92\Delta x^3$ and shown in Fig. R1B. The polynomial constitutive law here takes a fully nonlinear form, both in compression and in tension, as opposed to the previous examples with a linear tension regime. The importance of this can be understood in the context that the flip in the sign of displacement before pulse reflection off the opposite end of the system suggests tensile behavior. The qualitative difference between the identified optimal force-displacement curve for the pulse shape transformation and peak transmitted kinetic energy minimization problems can be seen, wherein the pulse shape transformation solution exhibits a relatively slight softening to stiffening behavior in compression and stiffening in tension.

Taking the DEM-identified nonlinearity as our target, we perform shape optimization to identify a unit cell shape to achieve this nonlinearity, this time using a neo-Hookean material model (under plane-strain conditions) to account for the large strains (up to 40%) in both directions, as seen in Fig. R1B. We note that unit cells undergoing geometric nonlinearity that were optimized using a similar process for different quasi-static nonlinearities have been previously realized from silicone confined in rigid plastic frames, and found to be in good agreement with predictions from the neo-Hookean model (see *e.g.*, Ref. [Xiu, H. et al. *Minimizing finite viscosity enhances relative kinetic energy absorption in bistable mechanical metamaterials* 596 but only with sufficiently fine discretization: a nonlinear dynamical size effect. *arXiv preprint arXiv:2410.02090 (2024)*]). The resultant optimized geometry is shown in Fig. R1C and the corresponding force-displacement relations obtained via FEM simulations, and validated with COMSOL, in Fig. R1B. We note that such a pulse transformation objective is similar to the formation of rarefaction waves in response to a compressive impact, which has previously been demonstrated in Ref. [Yasuda, H. et al. *Origami-based impact mitigation via rarefaction solitary wave creation. Sci. Advances* 5, eaau2835 (2019)], wherein qualitatively analogous softening-in-compression was used. We suggest that this similar result is of particular interest for two reasons: i) Our resulting identified microstructure unit cell geometry is qualitatively different than the origami solution (*e.g.*, our identified solution is in 2D instead of 3D, and not involving creased origami joints), and ii) Our optimization method found a force-displacement response with somewhat qualitatively overlapping features (initial softening in compression) from a random initial guess. We believe the latter point signifies the future potential for our method to identify previously unknown nonlinear dynamical mechanisms, such as was demonstrated by the optimum nonlinearity identified in the first problem, shown in Fig. 2.”

Figure R1: **Application of our method to a second problem: Pulse shape transformation.** (A) Input (dashed black) and target output (dashed purple) half-cycle sine waves, and the output waves obtained from DEM simulations with linear (solid black) and optimized (solid purple) nonlinear springs. (B) Force-displacement relations for the pulse shaping shown in (A), in contrast with the previous target of KE minimization, with additional fitted polynomial and secondary, validation, FEM (COMSOL) response. (C) Mesostructure shape identified by the optimizer to match the DEM-identified nonlinearity.

For instance, why is a nonlinear polynomial force-displacement relationship assumed? Why does this relationship include quadratic and cubic terms? Further

A nonlinear polynomial force-displacement relationship is assumed because polynomials provide a flexible and general way to approximate a wide variety of nonlinear behaviors. According to the Weierstrass approximation theorem, any continuous function within a certain range can be approximated by a polynomial to a desired degree of accuracy. Polynomials are relatively simple to handle both analytically and numerically (in contrast to, *e.g.*, non-differentiable piecewise functions), allowing for straightforward differentiation and integration, which is useful in dynamics analysis as well as optimization methods. A polynomial expression also captures a broad qualitative range of nonlinearities (*e.g.*, hardening, softening, non-monotonic, and symmetric and asymmetric behavior). Further, we found that inclusion of up to

fifth order polynomial terms resulted in minimal performance improvements compared to the third order representation for our primary case study of peak transmitted kinetic energy minimization.

We have clarified this point in the revised manuscript, as follows: “The nonlinear spring consists of an up-to-third-order polynomial, where the linear stiffness remains fixed at c_1^* . The non-dimensional nonlinear spring force is expressed as

$$f(\Delta x) = \Delta x + c_2(\Delta x)^2 + c_3(\Delta x)^3, \quad (\text{R.1})$$

where $\Delta x = \Delta x^*/a$ is the dimensionless spring stretch (with Δx^* the spring stretch defined such that elongation is positive), and c_2 and c_3 are dimensionless nonlinear coefficients of the second and third order-terms (with $c_n = c_n^* a^{n-1}/c_1^*$). We choose to describe the nonlinear springs as up-to-third-order polynomials due to the flexibility of this representation, namely, the ease with which they can represent a wide qualitative range of nonlinearities and the ease which polynomials lend to accurate dynamical simulation (as opposed to non-differentiable, *e.g.*, piecewise continuous functions). Along these lines, we found that inclusion of up to fifth order polynomial terms resulted in minimal performance improvements compared to the third order representation for our primary case study of peak transmitted kinetic energy minimization (see Supplementary Information Note 2).”

“SI—Note 2: Fifth order springs

Here we consider a nonlinear springs with a fifth order polynomial to govern its force-displacement relation, $f(\Delta x) = \Delta x + c_2(\Delta x)^2 + c_3(\Delta x)^3 + c_4(\Delta x)^4 + c_5(\Delta x)^5$. Considering deformations in the range $\Delta x \in [0, 1]$, the positive strain energy condition becomes

$$P(\Delta x) = \int_0^1 f(\Delta x) d\Delta x > 0, \quad (\text{R.2})$$

which leads to a constraint on the polynomial coefficients

$$30 + 20c_2 + 15c_3 + 12c_4 + 10c_5 > 0. \quad (\text{R.3})$$

The material system is simulated via DEM with $N = 20$ particles in the chain, with damping $\zeta = 0.01$ and with the impact conditions $M/M_0 = 0.05$ and $V/V_0 = 1$. For the optimization, the objective is chosen as the peak kinetic energy at the last particle, normalized by the linear response, *i.e.*, $\max(KE_{non})/\max(KE_{lin})$. A gradient based optimizer is used to minimize this objective, while constraining the design space to satisfy the positive energy condition, Eq. R.3. The best nonlinearity obtained from the optimization is $f(\Delta x) = \Delta x - 4.71\Delta x^2 + 1.97\Delta x^3 + 17.74\Delta x^4 - 15.05\Delta x^5$, which results in a performance of $\max(KE_{non})/\max(KE_{lin}) = 3.8\%$. The resultant optimized spring and the corresponding spatiotemporal KE response are shown in Fig. R2. The fifth order optimized springs display qualitative and quantitative similarity to the cubic polynomial springs discussed in the main text within the current deformation range; both having snap-through mechanisms and achieving comparable dynamic performances, thus demonstrating the adequacy of cubic polynomials for KE minimization.”

Figure R2: Results of optimized 5th order springs. (a) Spatiotemporal kinetic energy response. (b) Force-displacement relations of springs.

Why is a nonsymmetric behavior adopted in tension and compression? These aspects should be addressed and justified from a mechanical point of view.

For the case study shown in the original manuscript and focused primarily upon in the revised version, namely, the minimization of transmitted peak kinetic energy in response to an impact, nonsymmetric behavior in tension and compression (specifically linear behavior in tension and nonlinear behavior in compression) was chosen because of the specific figure of merit used. Under such loading conditions, significantly larger strain (which affects the chosen figure of merit) is experienced in compression compared to tension.

We have addressed this in the revised manuscript via the following:

1. Added text: “In this section, we describe the identification of an optimal nonlinear constitutive law for the case study of minimizing transmitted peak kinetic energy in response to an impact. Specifically, we simulate the impact of a rigid, variable mass and velocity rigid “impactor” incident on the left end of the chain. The model also incorporates a contact spring designed to facilitate the smooth contact and controlled release of the impactor during initial impact and rebound, respectively. Before searching for optimal nonlinear constitutive responses with our DEM, we set several bounds. First, for simplicity, we set the tension response to purely linear. In Supplementary Information Note 4, we show that the inclusion of nonlinearity in tension has little effect on the identified optima, which is as expected, due to the compressive nature of the impact event simulated (the identified optimum has a maximum compressive strain over five times greater than the maximum tensile strain).”

As per the above added text, in the revised Supplementary Information, we also added an optimization for the same impact condition and metric as in Fig. 2, but did not impose that the response would be linear in tension. Instead, it was described by the chosen polynomial throughout the entire range. It can be seen that the resulting KE ratio is 3.8%, whereas the best performing case in Fig. 2, was 3.98%. As such, for the metric of minimization of peak kinetic energy in response to impact, the use of linearity in tension has minimal effect. The added Supplementary Information section and Figures (Fig. R3 and R4) is shown below.

“SI—Note 4: Fully nonlinear springs

In this section we consider a fully nonlinear spring, which is governed by a third-order polynomial in both tensile and compressive deformations, as opposed to the linear tensile regime in the main text. The force-displacement relationship has the usual form $f(\Delta x) = \Delta x + c_2(\Delta x)^2 + c_3(\Delta x)^3$ and we consider deformations $\Delta x \in [-1, 1]$. To satisfy the positive strain energy for tensile and compressive deformations, we have

$$P^+(\Delta x) = \int_0^1 f(\Delta x)d\Delta x > 0 \quad \text{and} \quad P^-(\Delta x) = \int_0^{-1} f(\Delta x)d\Delta x > 0, \quad (\text{R.4})$$

which results in two constraints on the coefficients

$$c_2 > -\frac{3}{2} - \frac{3c_3}{4} \quad \text{and} \quad c_2 < \frac{3}{2} + \frac{3c_3}{4}. \quad (\text{R.5})$$

We conduct DEM simulations on the same system that is described in the main text, with $N = 20$ particles in the chain and $\zeta = 0.01$ and with the impact conditions $M/M_0 = 0.05$ and $V/V_0 = 1$. To investigate the effect of fully nonlinear springs, we start by a preliminary exploration of the parameter space $c_2 \in [-20, 10]$ and $c_3 \in [0, 100]$. The results are reported in terms of the maximum kinetic energy at the end of the material, normalized by the corresponding linear system, as shown in Fig. R3(a), followed by a finer search in the sensitive region shown in Fig. R3(b).

Figure R3: Normalized maximum kinetic energy at the last particle of the chain a function of nonlinear spring coefficients with the fully nonlinear springs. (a) Initial design space. (b) Sensitive region.

The gradient based optimization approach is applied to the problem to find the optimal spring coefficients by minimizing the KE ratio with using the constraints given in Eq. (R.5) and limiting the design space into the sensitive region in Fig. R3(b). The best spring is found to be $f(\Delta x) = \Delta x - 5.70\Delta x^2 + 10.54\Delta x^3$ with a KE ratio of $\max(KE_{non})/\max(KE_{lin}) = 3.8\%$, which is comparable to the results discussed in the main text. The resultant KE transmission is shown in Fig. R4(a) with the corresponding optimal spring in Fig. R4(b). We conclude that we have not observed a significant difference between using a fully nonlinear spring versus the one with a linear tensile regime in the context of KE minimization.”

2. We added markers which denote the maximum strains experienced in both tension and compression for the four cases shown in the revised Fig. 2B (Fig. R5, below).

Figure R4: Results of optimized fully nonlinear springs. (a) Spatiotemporal kinetic energy response. (b) Force displacement relations of springs.

Moreover, while a partial interpretation of these choices is provided from a static perspective, an analysis of the nonlinear interaction and explanation of the associated phenomena should be provided. The results presented in Fig. 2 are valuable and demonstrate the efficacy of the performed optimization. A description of the role of the parameters is provided with reference to the static force-displacement relationship. Additionally, the sensitivity of the response as a function of the parameter is analyzed, but a description of the dynamical behavior is missing.

We thank the Reviewer for highlighting the value of these aspects of our work, and have striven in the revised manuscript to provide additional analysis of the nonlinear dynamical interaction and associated dynamical phenomena. Specifically, we have revised Fig. 2 in the main text (Fig. R5) and added an entirely new Figure in the revised manuscript (Fig. R7), which we will use to address each specific related question, below.

What happens to the kinetic energy from cells 1 to 15? What mechanism stops the propagation of the mechanical waves? Is there a localization phenomenon?

We have addressed this in the revised manuscript via the following:

“A further important point of note about our identified optimum is that it is not of the form one would expect based on the conventional design approach for bistable energy absorption at lower rates [Shan, S. et al. Multistable architected materials for trapping elastic strain energy. *Adv. Mater.* 27, 4296–4301 (2015)], where net positive energy is locked into strain energy when snapping from the undeformed state to its second stable equilibrium (*i.e.*, where for energy locking, the area under the curve from $\Delta x = 0$ to the unstable equilibrium point is greater than the area under the curve from the unstable equilibrium to the second stable equilibrium point). In the case of our optimum, the response exhibits snap-through, but achieves neither bistability nor the more restrictive case of energy locking. Using this qualitatively different nonlinearity, our optimum outperforms the energy locking bistable case by over a factor of three (300% change via the above metric).

We next seek to understand *why* our optimum performs better than the other cases. Spatiotemporal responses of kinetic energy of the optimal nonlinear, underperforming nonlinear, and linear chain are shown in Fig. R5D-F, respectively. Most notably, while all materials in Fig. R5D-F see a pulse of energy propagating across the material, the pulse in the best performing case (Fig. R5D) appears to stop before

Figure R5: **Identification of optimal nonlinear constitutive response via DEM simulation for a single impact condition for the case of peak transmitted kinetic energy minimization.** (A) Feasible solutions of nonlinear spring coefficients c_2 and c_3 . The black line represents $c_2 = -\sqrt{3}c_3$, the red line indicates $c_2 = -(1 + 3c_3)/2$, the blue line is $c_2 = -3c_3$, and the green line is the zero strain energy throughout the whole range, $c_2 = -3/2 - 3c_3/4$. (B) Non-dimensional force-extension relationship of the best performing nonlinear spring ($f(\Delta x) = \Delta x + 5.88\Delta x^2 + 9.65\Delta x^3$) along with an example of a nearby underperforming (bad) nonlinear spring of $f(\Delta x) = \Delta x - 6.6\Delta x^2 + 11\Delta x^3$ (blue) and an energy locking bistable spring of $f(\Delta x) = \Delta x - 5.26\Delta x^2 + 6.75\Delta x^3$ (magenta). These three cases result in KE ratios of 0.0398 (best), 1.2257 (bad), and 0.1448 (energy locking bistable). (C) Ratio of maximum kinetic energy of the nonlinear spring to the one of a linear spring at the last particle as a function of nonlinear spring coefficients for the impact condition of $M/M_0 = 0.05$ and $V/V_0 = 1$, with $\zeta = 0.01$. The lines from (A) are overlaid, the star marker denotes the point of best performance, the triangle indicates the nearby case, and the square represents the bistable case. Normalized kinetic energy of the (D) best performing nonlinear, (E) underperforming (bad) nonlinear, and (F) linear material.

the end of the chain. Via the metric of minimizing peak kinetic energy at the end of the chain, it is clear why this case performs better. We next make several observations about the spatiotemporal responses. First, the traveling pulses in the two nonlinear cases (Fig. R5D,E) are more localized than that of the linear case (Fig. R5F). This is to be expected, due to the known formation of solitary waves (which localize via a balance of nonlinear and dispersive effects) in systems with qualitatively similar nonlinearities (snap-through) [Katz, S. and Givli, S. *Boomerons in a 1-d lattice with only nearest-neighbor interactions. Europhys. Lett. 131, 64002 (2020)*]. Indeed, in Ref. [Katz, S. and Givli, S. *Boomerons in a 1-d lattice with only nearest-neighbor interactions. Europhys. Lett. 131, 64002 (2020)*], they show that these solitary waves take the form of “boomerons”, where the wave arrest (and in their case reversal of direction) arises without any dissipation, and is suggested to be “a consequence of the intriguing interaction between the localized phenomenon and the trail of nonlinear waves”. In the Supplementary Information Note 6 we show spatial profiles of the pulses in our systems at several times to further visualize the observed localization phenomena.”

SI—Note 6: Self-localization in the optimal nonlinear system for minimizing peak transmitted kinetic energy

In Fig. R6, we examine the pulse profiles in our system at different time intervals to illustrate the localization phenomena occurring in the nonlinear, optimal system. For the linear system (Fig. R6(b)), the pulse exhibits a broader width, with an oscillating tail that grows in time, which is consistent with linear dispersive effects. In contrast, the pulses in the nonlinear material exhibit narrower widths and approximately retain their shape as it propagates, which is consistent with solitary wave behavior. A slight decay can be seen for both the linear and nonlinear systems, which can be attributed to the small level of damping present in the system, and in the linear case, dispersive effects (assuming the nonlinearity is balancing the dispersion in the nonlinear case).

Figure R6: Spatial profiles of solitary waves at different dimensionless time steps in the (a) linear and (b) nonlinear systems from Fig. 2D and 2F.

Is the energy transferred back to the source? A proper dynamical analysis of these aspects, supported by results (e.g., time response of the oscillator, analysis of the frequency content, etc.), is missing.

We have addressed this in the revised manuscript as follows:

“However, we suggest that boomeran-related wave arrest found in conservative systems is not the only mechanism contributing to the identified optimum performance. In Fig. R7A,B, we show the two-dimensional (2D) Fourier transforms of the normalized velocity of the optimal nonlinear (A) and linear (B) systems. In the linear system (Fig. R7B), the energy distribution can be seen to follow the expected dispersion of a monoatomic chain (a single “acoustic” branch, followed by stop band above a cutoff frequency [Brillouin, *L. Wave propagation in periodic structures*. McGraw-Hill Book Co. google schola 2, 2022–2025 (1946)]). In Fig. R7A, due to frequency conversion induced by the system’s nonlinearity, the energy is spread out to a much broader frequency range. This spreading has two effects. First, it converts some of the wave energy into the non-propagating stop band region, and second, it generates higher frequencies which more strongly activate energy loss via the inter-site damper (due to its proportionality with velocity). This latter effect can be seen in Fig. R7C, wherein the total energy of the entire chain decreases more quickly in the two nonlinear systems (both the best and the bad cases) than the linear system. This suggests that the ideal use of the underlying mechanisms for our peak transmitted kinetic energy minimization metric would thus be to decrease the total energy of the system via nonlinear-dissipation interplay,

while simultaneously arresting the propagating pulse before the end of the chain via the aforementioned boomeran mechanism. Such interplay between dissipation, damping, and wave arrest is consistent with the recent observation in bistable nonlinear systems [Fancher, R. et al. Dependence of the kinetic energy absorption capacity of bistable mechanical metamaterials on impactor mass and velocity. *Extrem. Mech. Lett.* 102044 (2023), Xiu, H. et al. *Minimizing finite viscosity enhances relative kinetic energy absorption in bistable mechanical metamaterials but only with sufficiently fine discretization: a nonlinear dynamical size effect.* *arXiv preprint arXiv:2410.02090* (2024)].”

Figure R7: **Contribution of the interaction of dissipation and nonlinearity to the minimization of transmitted peak kinetic energy.** Fourier transforms (dimensionless frequency vs. wavenumber) of the spatiotemporal normalized velocity of the optimal nonlinear (A) and linear (B) systems (corresponding to the spatiotemporal response of Fig. R5D and F, respectively). (C) Time evolution of total energy in the system, normalized by initial total energy (impactor kinetic energy) for three cases mentioned above.

Can the optimization create a band gap for wave propagation? Is it possible to extract a dispersion curve that clarifies the wave-stopping phenomenon? These questions should be clearly addressed considering the topic and the claims made.

It is indeed possible to use optimization to create a band gap in a mechanical metamaterial. Works such as Ref. [Sigmund and Jensen. Systematic design of phononic band-gap materials and structures by topology optimization. *Philosophical Transactions of the Royal Society of London. Series A: Mathematical, Physical and Engineering Sciences* 361.1806 (2003)] have explored targeted acoustic properties such as band gaps and wave guides using topology optimization. However, we emphasize that our presented wave manipulation phenomena cannot be encompassed via a dispersion relation, which is a fundamentally linear concept. Because the linear properties of both the linear and nonlinear systems are the same (they share the same linear stiffness), they share the same dispersion relation – that of a monoatomic spring-mass chain. We have addressed this as per the added text cited in response to the prior question, and in the revised introduction, as follows: “Focusing on the problem of minimizing peak kinetic energy transmitted via waves in response to impact, we first note that because impact is inherently a broadband excitation, prior linear mesostructured material (or “metamaterial”) strategies that leverage bandgaps [Hussein, M. I., Leamy, M. J. and Ruzzene, M. *Dynamics of phononic materials and structures: Historical origins, recent progress, and future outlook.* *Appl. Mech. Rev.* 66, 040802 (2014)] have been shown to have limited efficacy (e.g., requiring gradient or disordered material strategies that increase bandwidth at the cost of attenuation [Tan, K. T., Huang, H. and Sun, C. *Blast-wave impact mitigation using negative effective mass density concept of elastic metamaterials.* *Int. J. Impact Eng.* 64, 20–29 (2014)]). In favor of this, several nonlinear mesostructured material motifs have been realized and studied in the context of impact ...”

Experimental Campaign

The performed activities have great experimental value.

We thank the Reviewer for highlighting the experimental value of our work.

However, a more accurate analysis of the observed response is missing. Considering the advanced experimental setup employed, it would have been useful to show the time history of the response to clarify the dynamical interactions.

We note that the time histories for all experiments conducted were shown in the original manuscript in the main text (Fig. 5A and C in the original manuscript) and the SI (all other trials) via spatiotemporal maps of the measured kinetic energy at each particle.

To clarify the time history of the system response, we have added to the Figure (previous Fig. 5, now Fig. 6 in the revised manuscript, and Fig. R8 in the response below), as panels E and F, the velocity time histories for particles (both simulation and experiment) 5 and 20.

Additionally, the following text has been added to the main text:

“Time histories of measured velocity at the fifth and the last unit cell can be seen in Fig. R8E-F, highlighting the greatly reduced last particle velocity in the nonlinear case as compared to the linear case.”

Figure R8: **Experimental validation of the optimal nonlinear and linear chains, compared with simulation.** The optimal nonlinear chain corresponds to that identified in Fig. 4A. (A-D) Spatiotemporal evolution of kinetic energy in the system (normalized by the input kinetic energy, or initial total energy). The nonlinear chain is shown in A (experiment, nonlinear trial 1) and B (simulation), while the linear case is shown in C (experiment, linear trial 1) and D (simulation). The experimental spatiotemporal plots include a smoothing of displacement values to assist with noise induced by differentiating the discrete time camera data. The spatiotemporal plots for other trials are included in Supplementary Information Note 9. (E-F) Time histories of the experimentally measured (solid lines) and simulated (dashed lines) velocities for the fifth and last unit cell in the nonlinear (E) and linear (F) chains. (G) A ratio of maximum kinetic energy (linear/nonlinear) seen at each unit cell for both experiment and simulation. A value greater than 1 (denoted by the horizontal dashed black line) indicates superior performance of the nonlinear chain as compared to the linear. The experiment values are the average taken from three experimental trials. The X marks the average experimental value recorded by the vibrometer (see Supplementary Information Note 9 for the full data sets), which collected data from only the last unit cell. (H) The simulated sensitivity of the KE ratio (truncated at 1) to impact conditions, wherein the simulation was run with the coefficients that were found experimentally from the nonlinear spring in Fig. R11D (*i.e.*, the physically achieved coefficients).

A discussion about the practical limitations of realizing this kind of nonlinear chain in a new material is important, considering that this seems to be the main problem to overcome.

Further discussion regarding the practical limitations has been added to the main text, as follows:

“As suggested by the physically large size of our experimentally realized system, we note that this class of geometrically-nonlinear-microstructure-enabled wave transforming materials demands a particularly chal-

lenging set of manufacturing requirements. This is due to a particularly large scale separation between the system size and the smallest features. For instance, the optimized design shown in Fig. R11B contains very thin hinges at the top and bottom of the central beam, particularly compared to the size of the unit cell. Given that the wave-dominated behavior shown herein requires many unit cells, we have a situation with three distinct separated length scales (smallest feature, unit cell, system). While this length scale separation is in some sense shared by other metamaterial and lattice structures [Cummer, S. A., Christensen, J. & Alu, A. *Controlling sound with acoustic metamaterials. Nat. Rev. Mater.* 1, 1–13 (2016), Jiao, P., Mueller, J., Raney, J. R., Zheng, X. & Alavi, A. H. *Mechanical metamaterials and beyond. Nat. communications* 14, 6004 (2023)], it is amplified for our case. For instance, at the unit cell level, reduction in this disparity is possible, however it comes at the expense of limiting the range of nonlinearity achievable (*i.e.*, one often needs long, thin, deformable structures to achieve large nonlinear response within elastic regimes). Future incorporation of plasticity, phase transformation, or contact may address this issue. Further, recent works have shown the existence of a nonlinear dynamic “size effect” where a critical, minimal number of unit cells has been shown to yield enhanced performance [Xiu, H. *et al. Minimizing finite viscosity enhances relative kinetic energy absorption in bistable mechanical metamaterials but only with sufficiently fine discretization: a nonlinear dynamical size effect. arXiv preprint arXiv:2410.02090* (2024)], in addition to the longer system size compared to the characteristic wavelengths giving more time and space for traveling pulses to evolve. Indeed, relatively coarse parameter sweeps of the cubic polynomial coefficients for the peak transmitted kinetic energy minimization problem for more highly discretized materials (*e.g.*, 100 unit cells, see Supplementary Information Note 7), suggest the potential for over two orders of magnitude improvement in kinetic energy transmission via the use of a nonlinear material compared to a comparative linear material. All of these length scale separation manufacturing challenges would be further exacerbated in 3D, with the addition of loss of the mechanism for tuning the linear stiffness via out-of-plane structure depth. However, we note that recent advances in 3D printing have seen large jumps in the system to smallest feature size ratio versus manufacturing speed [Kiefer, P. *et al. A multi-photon (7×7)-focus 3d laser printer based on a 3d-printed diffractive optical element and a 3d-printed multi-lens array. Light. Adv. Manuf.* 4, 28–41 (2024)]. Self-assembly-based manufacturing is a further tantalizing possibility [Jin, H. & Espinosa, H. D. *Mechanical metamaterials fabricated from self-assembly: A perspective. J. Appl. Mech.* 91 (2024)].”

In conclusion, it is the reviewer’s opinion that the paper cannot be published in its current form because the main claim is not properly addressed. The content is interesting and relevant but is more focused on the different tasks performed. A broader perspective, from which general and structured findings emerge, is missing.

We thank the reviewer for their feedback, and, as per the added and revised sections described in the above responses, have attempted to make sure our main claim was clarified and properly addressed, and that a broader perspective provided. As described in the revised introduction, our main claims are:

- The introduction of a method that enables the creation of “customizable wave tailoring materials via nonlinear bilevel inverse design”
- That this is a broadly applicable method, which we demonstrate via the application of “it to two problems—minimization of peak transmitted kinetic energy in response to an impact, and pulse shape transformation (inversion of an applied boundary displacement signal at the other end of the chain).”
- That within the context of impact mitigation with wave tailoring materials, that we obtained two key “fundamental insights: i) Small differences in the nonlinearity can drastically change the dynamic response of the system, from severely under- to outperforming a comparative linear system;

and ii) The often used design strategy of impact mitigation via “energy locking” bistability can be significantly outperformed by our identified optimal nonlinearity.”

With the addition of the rewritten abstract and introduction, and the further explorations of the mechanisms by which kinetic energy transmission is mitigated, the effect of including additional nonlinear terms, and the application the developed framework to different potential applications, we hope that we have addressed reviewer’s concerns. In addition, we have added a drastically expanded discussion section, which, in addition to the paragraph cited in response to the prior question, is as follows.

“While we have demonstrated our design method on a relatively simple (1D, elastic regime, large, few unit cell) system, and extracted several new fundamental insights for our primary case study of peak transmitted kinetic energy minimization, we suggest this sets a foundation for future research and technological development. Most directly, within the considered case study of impact mitigation, a near term question of future interest would be how such optimal nonlinearities would change for a different metric such as maximum transmitted force or peak tensile stress anywhere in the material. Similarly immediate, given the flexibility of a designer nonlinear spring and the incorporation into a lumped-mass context shown herein, we suggest this enables the possibility for rapid experimental investigation of nonlinear dynamical phenomena only seen thus far in simulation and theory. Less directly, one can imagine extending the bilevel optimization to 3D and to more complex constituent material models. The extension to 3D yields particularly non-trivial challenges, due to the second-order tensorial nature of solid mechanical systems, including with the potential for shear-to-longitudinal mode conversion in linear [*Graff, K. F. Wave motion in elastic solids (Courier Corporation, 2012)*], and not-to-mention-nonlinear [*Wallen, S. P. & Boechler, N. Shear to longitudinal mode conversion via second harmonic generation in a two-dimensional microscale granular crystal. Wave motion 68, 22–30 (2017)*], dynamical regimes. With regards to optimization of unit cells in 3D for desired quasi-static nonlinear mechanical response, some initial steps have recently been made for select multiaxial loading states [*Zheng, L., Kochmann, D. M. & Kumar, S. Hypercan: Hypernetwork-driven deep parameterized constitutive models for metamaterials. Extrem. Mech. Lett. 72, 102243 (2024)*]. Extension of this method to less scale separated dynamical (no lumped masses) would enable more mass efficient structures, but would likely required either some degree of homogenization or larger, high-performance capability. The incorporation of irreversibility of the constituent material model, such as plasticity, would open enhanced energy dissipation mechanisms and application to high strain-rate applications [*Meyers, M. A. Dynamic behavior of materials (John wiley & sons, 1994)*]. Similarly, the incorporation of activity or stimuli-responsivity [*Xia, X., Spadaccini, C. M. & Greer, J. R. Responsive materials architected in space and time. Nat. Rev. Mater. 7, 683–701 (2022)*] in the constituent material model would open applications in shape morphing, mechanical computing, and the capacity for materials that autonomously respond and conduct directed work [*Jiao, P., Mueller, J., Raney, J. R., Zheng, X. & Alavi, A. H. Mechanical metamaterials and beyond. Nat. communications 14, 6004 (2023)*]. The use of shape memory materials [*Xia, X., Spadaccini, C. M. & Greer, J. R. Responsive materials architected in space and time. Nat. Rev. Mater. 7, 683–701 (2022)*] offers potential in each of these capacities, via targeted shape change, potential resetting capability, and enhanced energy absorption. Incorporation of multiphysics coupling into the material model may similarly open up new possibilities in fields such as optomechanics [*Aspelmeyer, M., Kippenberg, T. J. & Marquardt, F. Cavity optomechanics. Rev. Mod. Phys. 86, 1391–1452 (2014)*].

In each of the above cases, in addition to opening new application possibilities, we believe that this method has the potential to help answer fundamental questions concerning the interplay of nonlinearity with these varied mechanisms, and most generally the spatiotemporal partition of energy in nonlinear dynamical systems [*Fermi, E., Pasta, P., Ulam, S. & Tsingou, M. Studies of the nonlinear problems. Tech. Rep., Los Alamos National Laboratory (LANL), Los Alamos, NM (United States) (1955)*]. Such nonlinear dynamical questions increase in complexity greatly, when considering the influence of disorder [*Pikovskiy, A.*

Shepelyansky, D. *Destruction of anderson localization by a weak nonlinearity. Phys. review letters* 100, 094101 (2008)], including in the form of designed defect placement or spatially extended heterogeneity. It also raises fundamental questions, and via this method a possible new avenue for answering, in the form of design of nonlinear systems in the context of stimuli sensitivity. For instance, in our case of peak transmitted kinetic energy minimization, we observed (as common for many nonlinear systems) large sensitivity of the kinetic energy transmission metric to small changes in nonlinearity and impact conditions. This indicates future possibilities where computational design can incorporate stimuli sensitivity and parameter uncertainty into the design objective. Further, one can also imagine the potential to leverage the tools developed in the nonlinear dynamical systems community. Instead of optimizing for features in the time domain response of the system, one may optimize in the context of bifurcation diagrams [Boechler, N. et al. *Discrete breathers in one-dimensional diatomic granular crystals. Phys. review letters* 104, 244302, Boechler, N., Theocharis, G. & Daraio, C. *Bifurcation-based acoustic switching and rectification. Nat. materials* 10, 665–668 (2011)] or phase space [Strogatz, S. H. *Nonlinear dynamics and chaos: with applications to physics, biology, chemistry, and engineering (CRC press, 2018)*], where in the case of the former, new modes emerge and disappear with variation of input parameters or change stability, and in the latter, nonlinear dynamical objects such as limit cycles and strange attractors are formed. Noting the capacity for systems such as ours to be described via partial differential equations and continuum approximations (most directly, such as the KdV description of the FPUT system [Dauxois, T. & Peyrard, M. *Physics of Solitons (Cambridge University Press, 2006)*]), analysis tools developed for these limits [Scott, A. *Encyclopedia of nonlinear science (Routledge, 2006)*] may be incorporated into optimization objectives to improve the ability for the computer to meaningfully traverse and efficiently search the design space. Considering all these possibilities together, results in a rich array of possible future extensions.”

Responses to Reviewer 2:

In this article, the authors describe a way in which the nonlinearity of a spring-mass system can be optimized to minimise the kinetic energy at one end of the spring-mass system given an initial ‘jolt’ to the system at the other end. This is achieved by the solution of a discrete model using numerical methods that is used to inform the design of a physical material in an experiment that also minimises the kinetic energy.

Firstly, I’d like to congratulate the authors on an really well-written article, I didn’t spot a single typo, it was written clearly, and, I believe after an appropriate amount of study on my behalf, to have understood what they are trying to achieve here. I’d like to emphasise that I see nothing wrong with their findings, they were extremely transparent with their methodology and I have no reason to doubt the validity of their work. I also found it really interesting, so well-done.

We thank the reviewer for their careful review and the positive comments regarding our work.

However, I do have some reservations with regards to whether this article may be better suited to a different journal, as I do not believe the authors have convinced me this is important enough for inclusion into Nature Comm, I think that perhaps Sci. Rep. may be a more appropriate place for it. Whilst I enjoyed the article and was impressed by the results, I was left with a feeling of ‘so-what?’ which I think the authors could’ve done a better job

of setting the future context.

We thank the Reviewer for this comment, as indeed the responsibility is on us as authors to convey the importance of our work to a general readership. Regarding this point, we have added further context, analysis, and attempted to more clearly convey the importance and future potential applications of our work. This includes a new second example to the minimization of peak transmitted kinetic energy to demonstrate the potential breadth of applicability of our method, namely, pulse shaping. The key revisions in this regard are copied below, which includes a rewritten abstract, introduction, discussion section, and the added example on pulse shaping.

Abstract: “Passive wave transformation via nonlinearity is ubiquitous in settings from acoustics to optics and electromagnetics. It is well known that different nonlinearities yield different effects on propagating signals, which raises the question of “what precise nonlinearity is the best for a given wave tailoring application?” Considering a one-dimensional spring-mass chain connected by polynomial springs (a variant of the celebrated Fermi-Pasta-Ulam-Tsingou system), we introduce a bilevel inverse design method which couples the shape optimization of structures for tailored constitutive responses with reduced-order nonlinear dynamical inverse design. We apply it to two qualitatively distinct problems—minimization of peak transmitted kinetic energy, and pulse shape transformation—demonstrating our method’s breadth of applicability. Focusing on the first problem, we obtain two fundamental insights: i) Small differences in the nonlinearity can drastically change the dynamic response of the system, from severely under- to outperforming a comparative linear system; and ii) The often used design strategy of impact mitigation via “energy locking” bistability can be significantly outperformed by our identified optimal nonlinearity. We validate this case with impact experiments and find excellent agreement. This work sets the foundation for broader passive nonlinear mechanical wave tailoring material design, with applications to computing, signal processing, shock mitigation, and autonomous materials.”

Introduction: “In addition to the introduction of this method, we apply it to two qualitatively distinct—minimization of peak transmitted kinetic energy in response to an impact, and pulse shape transformation (inversion of an applied boundary displacement signal at the other end of the chain)—demonstrating the potential breadth of applicability of our method. We highlight also that in both cases, we conduct a comparison between the linear and nonlinear response. We assert that this comparison is particularly important from a fundamental perspective, as it isolates the role of nonlinearity from other linear wave manipulating effects such as dispersion, dissipation, and heterogeneity.

Focusing on the problem of minimizing peak kinetic energy transmitted via waves in response to impact, we first note that because impact is inherently a broadband excitation, prior linear mesostructured material (or “metamaterial”) strategies that leverage bandgaps [*Hussein, M. I., Leamy, M. J. and Ruzzene, M. Dynamics of phononic materials and structures: Historical origins, recent progress, and future outlook. Appl. Mech. Rev. 66, 040802 (2014)*] have been shown to have limited efficacy (*e.g.*, requiring gradient or disordered material strategies that increase bandwidth at the cost of attenuation [*Tan, K. T., Huang, H. and Sun, C. Blast-wave impact mitigation using negative effective mass density concept of elastic metamaterials. Int. J. Impact Eng. 64, 20–29 (2014)*]). In favor of this, several nonlinear mesostructured material motifs have been realized and studied in the context of impact, leveraging nonlinearities such as the aforementioned contact (tensionless and stiffening in compression [*Nesterenko, V. Dynamics of heterogeneous materials (Springer Science and Business Media, 2013)*]), tensegrity (stiffening in tension and tunably softening or stiffening in compression [*Fraternali, F., Carpentieri, G., Amendola, A., Skelton, R. E. and Nesterenko, V. F. Multiscale tunability of solitary wave dynamics in tensegrity metamaterials. Appl. Phys. Lett. 105, 201903 (2014)*]), and compressed beams (bistability and snap-through in compression [*Shan, S. et al. Multistable architected materials for trapping elastic strain energy. Adv. Mater. 27, 4296–4301 (2015)*]), as well as o-rings (tensionless with double-power law response in compression [*Herbold, E. and Nesterenko, V. Solitary and shock waves in discrete strongly nonlinear double power-law materials.*

Appl. Phys. Lett. 90 (2007)) and origami (softening in compression and stiffening in tension [*Yasuda, H. et al. Origami-based impact mitigation via rarefaction solitary wave creation. Sci. Advances* 5, eaau2835 (2019)]). Herein, using our method, we obtain the following two new fundamental insights within this area. The first, is that very small differences in the nonlinear spring response (without qualitative difference) can drastically change the response of the system (changing the nonlinear coefficients by less than 14% results in an over 3000% change in performance) in such dynamical settings, from severely under- to outperforming a comparative linear system. To our understanding, this degree of sensitivity was hitherto unknown in the context of impact mitigation. The second, is that the often used design strategy of impact mitigation via “energy locking” bistability [*Shan, S. et al. Multistable architected materials for trapping elastic strain energy. Adv. Mater.* 27, 4296–4301 (2015) and *Fancher, R. et al. Dependence of the kinetic energy absorption capacity of bistable mechanical metamaterials on impactor mass and velocity. Extrem. Mech. Lett.* 102044 (2023)] can be greatly outperformed by our identified optimal snap-through nonlinearity (by over a factor of three times). We then choose a second high performing identified nonlinear constitutive response (which is more amenable to our experimental capabilities) and demonstrate the full inverse design of a superior impact mitigation system, from identification of an ideal nonlinearity, to the design of the unit cell geometry, and experimental validation of the performance. While we focus the majority of the manuscript on this impact mitigation problem, as demonstrated by our application of this method to the aforementioned two qualitatively different objectives, we believe our method will have wide future applicability, including to the previously referenced array of nonlinear acoustic, phononic, and mechanical wave transformation applications [...]

Discussion: “As suggested by the physically large size of our experimentally realized system, we note that this class of geometrically-nonlinear-microstructure-enabled wave transforming materials demands a particularly challenging set of manufacturing requirements. This is due to a particularly large scale separation between the system size and the smallest features. For instance, the optimized design shown in Fig. R11B contains very thin hinges at the top and bottom of the central beam, particularly compared to the size of the unit cell. Given that the wave-dominated behavior shown herein requires many unit cells, we have a situation with three distinct separated length scales (smallest feature, unit cell, system). While this length scale separation is in some sense shared by other metamaterial and lattice structures [*Cummer, S. A., Christensen, J. & Alu, A. Controlling sound with acoustic metamaterials. Nat. Rev. Mater.* 1, 1–13 (2016), *Jiao, P., Mueller, J., Raney, J. R., Zheng, X. & Alavi, A. H. Mechanical metamaterials and beyond. Nat. communications* 14, 6004 (2023)], it is amplified for our case. For instance, at the unit cell level, reduction in this disparity is possible, however it comes at the expense of limiting the range of nonlinearity achievable (*i.e.* one often needs long, thin, deformable structures to achieve large nonlinear response within elastic regimes). Future incorporation of plasticity, phase transformation, or contact may address this issue. Further, recent works have shown the existence of a nonlinear dynamic “size effect” where a critical, minimal number of unit cells has been shown to yield enhanced performance [*Xiu, H. et al. Minimizing finite viscosity enhances relative kinetic energy absorption in bistable mechanical metamaterials but only with sufficiently fine discretization: a nonlinear dynamical size effect. arXiv preprint arXiv:2410.02090 (2024)*], in addition to the longer system size compared to the characteristic wavelengths giving more time and space for traveling pulses to evolve. Indeed, relatively coarse parameter sweeps of the cubic polynomial coefficients for the peak transmitted kinetic energy minimization problem for more highly discretized materials (*e.g.*, 100 unit cells, see Supplementary Information Note 7), suggest the potential for over two orders of magnitude improvement in kinetic energy transmission via the use of a nonlinear material compared to a comparative linear material. All of these length scale separation manufacturing challenges would be further exacerbated in 3D, with the addition of loss of the mechanism for tuning the linear stiffness via out-of-plane structure depth. However, we note that recent advances in 3D printing have seen large jumps in the system to smallest feature size ratio versus manufacturing speed [*Kiefer, P. et al. A multi-photon (7×7)-focus 3d laser printer based on*

a 3d-printed diffractive optical element and a 3d-printed multi-lens array. *Light. Adv. Manuf.* 4, 28–41 (2024)]. Self-assembly-based manufacturing is a further tantalizing possibility [Jin, H. & Espinosa, H. D. *Mechanical metamaterials fabricated from self-assembly: A perspective.* *J. Appl. Mech.* 91 (2024)].

While we have demonstrated our design method on a relatively simple (1D, elastic regime, large, few unit cell) system, and extracted several new fundamental insights for our primary case study of peak transmitted kinetic energy minimization, we suggest this sets a foundation for future research and technological development. Most directly, within the primary case study of impact mitigation, a near term question of future interest would be how such optimal nonlinearities would change for a different metric such as maximum transmitted force or peak tensile stress anywhere in the material. Similarly immediate, given the flexibility of a designer nonlinear spring and the incorporation into a lumped-mass context shown herein, we suggest this enables the possibility for rapid experimental investigation of nonlinear dynamical phenomena only seen thus far in simulation and theory. Less directly, one can imagine extending the bilevel optimization to 3D and to more complex constituent material models. The extension to 3D yields particularly non-trivial challenges, due to the second-order tensorial nature of solid mechanical systems, including with the potential for shear-to-longitudinal mode conversion in linear [Graff, K. F. *Wave motion in elastic solids* (Courier Corporation, 2012)], and not-to-mention nonlinear [Wallen, S. P. and Boechler, N. *Shear to longitudinal mode conversion via second harmonic generation in a two-dimensional microscale granular crystal.* *Wave motion* 68, 22–30 (2017)], dynamical regimes. With regards to optimization of unit cells in 3D for desired quasi-static nonlinear mechanical response, some initial steps have recently been made for select multiaxial loading states [Zheng, L. et al., *Hypercan: Hypernetwork-driven deep parameterized constitutive models for metamaterials.* *Extrem. Mech. Lett.* 72, 102243 (2024)]. Extension of this method to less scale separated dynamical (no lumped masses) would enable more mass efficient structures, but would likely required either some degree of homogenization or larger, high-performance capability. The incorporation of irreversibility of the constituent material model, such as plasticity, would open enhanced energy dissipation mechanisms and application to high strain-rate applications [Meyers, M. A. *Dynamic behavior of materials* (John Wiley and Sons, 1994)]. Similarly, the incorporation of activity or stimuli-responsivity [Xia, X. et al. *Responsive materials architected in space and time.* *Nat. Rev. Mater.* 7, 683–701 (2022)] in the constituent material model would open applications in shape morphing, mechanical computing, and the capacity for materials that autonomously respond and conduct directed work [Jiao, P. et al. *Mechanical metamaterials and beyond.* *Nat. communications* 14, 6004 (2023)]. The use of shape memory materials [Xia, X. et al. *Responsive materials architected in space and time.* *Nat. Rev. Mater.* 7, 683–701 (2022)] offers potential in each of these capacities, via targeted shape change, potential resetting capability, and enhanced energy absorption. Incorporation of multiphysics coupling into the material model may similarly open up new possibilities in fields such as optomechanics [Aspelmeyer, M. et al. *Cavity optomechanics.* *Rev. Mod. Phys.* 86, 1391–1452 (2014)].

In each of the above cases, in addition to opening new application possibilities, we believe that this method has the potential to help answer fundamental questions concerning the interplay of nonlinearity with these varied mechanisms, and most generally the spatiotemporal partition of energy in nonlinear dynamical systems [Fermi, E., Pasta, P., Ulam, S. and Tsingou, M. *Studies of the nonlinear problems.* *Tech. Rep., Los Alamos National Laboratory (LANL), Los Alamos, NM (United States) (1955)*]. Such nonlinear dynamical questions increase in complexity greatly, when considering the influence of disorder [Pikovsky, A. and Shepelyansky, D. *Destruction of anderson localization by a weak nonlinearity.* *Phys. review letters* 100, 094101 (2008)], including in the form of designed defect placement or spatially extended heterogeneity. It also raises fundamental questions, and via this method a possible new avenue for answering, in the form of design of nonlinear systems in the context of stimuli sensitivity. For instance, in our case of peak transmitted kinetic energy minimization, we observed (as common for many nonlinear systems) large sensitivity of the kinetic energy transmission metric to small changes in nonlinearity and impact conditions. This indicates future possibilities where computational design can incorporate stimuli sensitivity and

parameter uncertainty into the design objective. Further, one can also imagine the potential to leverage the tools developed in the nonlinear dynamical systems community. Instead of optimizing for features in the time domain response of the system, one could imagine optimizing in the context of bifurcation diagrams [Boechler, N. et al. *Bifurcation-based acoustic switching and rectification. Nat. materials* 10,665–668 (2011)] or phase space [Strogatz, S. H. *Nonlinear dynamics and chaos: with applications to physics, biology, chemistry, and engineering (CRC press, 2018)*], where in the case of the former, new modes emerge and disappear with variation of input parameters or change stability, and in the latter, nonlinear dynamical objects such as limit cycles and strange attractors are formed. Noting the capacity for systems such as ours to be described via partial differential equations and continuum approximations (most directly, such as the KdV description of the FPUT system [Dauxois, T. and Peyrard, M. *Physics of Solitons (Cambridge University Press, 2006)*]), analysis tools developed for these limits [Scott, A. *Encyclopedia of nonlinear science (Routledge, 2006)*] may be incorporated into optimization objectives to improve the ability for the computer to meaningfully traverse of the design space. Considering all these possibilities together, results in a rich, if not overwhelming, array of possible future extensions.”

New section—Application of method to a second problem: Pulse shape transformation.

“The bilevel inverse design approach can be applied to different problems of interest by altering the dynamic objective and finding the corresponding optimal nonlinearity. As such, to demonstrate the potential breadth of applicability of our model, we pursue a second problem: passive signal transformation, wherein we supply an input pulse to the material and seek to transform it into a desired shape at the opposite side. We perturb the system at the left boundary via a prescribed half-cycle sine wave displacement and seek to flip the sign of the pulse at the opposite end of the material, but maintain the same period and amplitude as the input, as shown with dashed lines in Fig. R9A. The optimal nonlinearity, identified using a gradient-based search algorithm (see Methods), which flips the input pulse as seen in Fig. R9A, is found to be $f(\Delta x) = \Delta x - 2.35\Delta x^2 + 4.92\Delta x^3$ and shown in Fig. R9B. The polynomial constitutive law here takes a fully nonlinear form, both in compression and in tension, as opposed to the previous examples with a linear tension regime. The importance of this can be understood in the context that the flip in the sign of displacement before pulse reflection off the opposite end of the system suggests tensile behavior. The qualitative difference between the identified optimal force-displacement curve for the pulse shape transformation and peak transmitted kinetic energy minimization problems can be seen, wherein the pulse shape transformation solution exhibits a relatively slight softening to stiffening behavior in compression and stiffening in tension.

Taking the DEM-identified nonlinearity as our target, we perform shape optimization to identify a unit cell shape to achieve this nonlinearity, this time using a neo-Hookean material model (under plane-strain conditions) to account for the large strains (up to 40%) in both directions, as seen in Fig. R9B. We note that unit cells undergoing geometric nonlinearity that were optimized using a similar process for different quasi-static nonlinearities have been previously realized from silicone confined in rigid plastic frames, and found to be in good agreement with predictions from the neo-Hookean model (see *e.g.*, Ref. [Xiu, H. et al. *Minimizing finite viscosity enhances relative kinetic energy absorption in bistable mechanical metamaterials but only with sufficiently fine discretization: a nonlinear dynamical size effect. arXiv preprint arXiv:2410.02090 (2024)*]). The resultant optimized geometry is shown in Fig. R9C and the corresponding force-displacement relations obtained via FEM simulations, and validated with COMSOL, in Fig. R9B. We note that such a pulse transformation objective is similar to the formation of rarefaction waves in response to a compressive impact, which has previously been demonstrated in Ref. [Yasuda, H. et al. *Origami-based impact mitigation via rarefaction solitary wave creation. Sci. Advances* 5, eaau2835 (2019)], wherein qualitatively analogous softening-in-compression was used. We suggest that this similar result is of particular interest for two reasons: i) Our resulting identified microstructure unit cell geometry is qualitatively different than the origami solution (*e.g.*, our identified solution is in 2D instead of 3D,

and not involving creased origami joints), and ii) Our optimization method found a force-displacement response with somewhat qualitatively overlapping features (initial softening in compression) from a random initial guess. We believe the latter point signifies the future potential for our method to identify previously unknown nonlinear dynamical mechanisms, such as was demonstrated by the optimum nonlinearity identified in the first problem, shown in Fig. 2.”

Figure R9: **Application of our method to a second problem: Pulse shape transformation.** (A) Input (dashed black) and target output (dashed purple) half-cycle sine waves, and the output waves obtained from DEM simulations with linear (solid black) and optimized (solid purple) nonlinear springs. (B) Force-displacement relations for the pulse shaping shown in (A), in contrast with the previous target of KE minimization, with additional fitted polynomial and secondary, validation, FEM (COMSOL) response. (C) Mesostructure shape identified by the optimizer to match the DEM-identified nonlinearity.

I have the following comments:

1) Equation (1) is an adaptation of the famous Fermi-Pasta-Ulam-Tsingou problem; I wonder if the authors missed a trick here by not mentioning/referencing such a famous problem (apologies if I missed it). Putting a fresh spin on such an important work in the history of nonlinear mathematics/physics would put this work in a better historical perspective. To that end, I wonder if equation (1) can be reduced to a KdV-type PDE when applying a continuum hypothesis; this would be valid for a large number of cells with a small gap-width and it may be an interesting future direction; could the authors comment on this? Have they considered the FPUT problem and how it links into their results?

Thank you for the reviewer’s valuable suggestion. We note that Equation (1) has the same form as the Fermi-Pasta-Ulam-Tsingou (FPUT) model with the addition of a viscous damping term. In the continuum limit and starting from Eq. (S2) in SI, the damped FPUT model can be expressed as:

$$m \frac{\partial^2 u}{\partial t^2} + \eta^* a^2 \frac{\partial^3 u}{\partial x^2 \partial t} = c_1^* a^2 \frac{\partial^2 u}{\partial x^2} + \frac{c_1^* a^4}{12} \frac{\partial^4 u}{\partial x^4} + 2c_2^* a^3 \frac{\partial u}{\partial x} \frac{\partial^2 u}{\partial x^2} + 3c_3^* a^4 \left(\frac{\partial u}{\partial x} \right)^2 \frac{\partial^2 u}{\partial x^2}, \quad (\text{R.6})$$

where $u(x, t)$ is a continuous function representing displacement as a function of space and time. The equation can be reformulated using the speed of sound $V_0 = a\sqrt{c_1^*/m}$:

$$\frac{1}{V_0^2} \frac{\partial^2 u}{\partial t^2} + \frac{2\zeta a}{V_0} \frac{\partial^3 u}{\partial x^2 \partial t} = \frac{\partial^2 u}{\partial x^2} + \frac{a^2}{12} \frac{\partial^4 u}{\partial x^4} + \frac{2c_2^* a}{c_1^*} \frac{\partial u}{\partial x} \frac{\partial^2 u}{\partial x^2} + \frac{3c_3^* a^2}{c_1^*} \left(\frac{\partial u}{\partial x} \right)^2 \frac{\partial^2 u}{\partial x^2}. \quad (\text{R.7})$$

This equation can be further simplified to the KdV PDE assuming the solution is slowly varying in both space and time. Let $\xi = x - V_0 t$, $\tau = (c_2^*/c_1^*)V_0 t$, and $\tilde{u}(\xi, \tau) = u(x, t)$. Under this change of coordinates, the equation becomes

$$\frac{c_2^* a}{2c_1^*} \frac{\partial^2 \tilde{u}}{\partial \tau^2} - \frac{\partial^2 \tilde{u}}{\partial \tau \partial \xi} + \zeta a \frac{\partial \tilde{u}}{\partial \tau} - \frac{\zeta c_1^*}{c_2^*} \frac{\partial^3 \tilde{u}}{\partial \xi^3} = \frac{c_1^* a}{24c_2^*} \frac{\partial^4 \tilde{u}}{\partial \xi^4} + \frac{\partial \tilde{u}}{\partial \xi} \frac{\partial^2 \tilde{u}}{\partial \xi^2} + \frac{3c_3^* a}{2c_2^*} \left(\frac{\partial \tilde{u}}{\partial \xi} \right)^2 \frac{\partial^2 \tilde{u}}{\partial \xi^2}. \quad (\text{R.8})$$

To take the continuum limit, assume that c_1^*a/c_2^* approaches a constant, while both c_2^*/c_1^* and a tend to zero. In this limit, we obtain:

$$\frac{\partial^2 \tilde{u}}{\partial \tau \partial \xi} + \frac{\zeta c_1^*}{c_2^*} \frac{\partial^3 \tilde{u}}{\partial \xi^3} + \frac{c_1^* a}{24 c_2^*} \frac{\partial^4 \tilde{u}}{\partial \xi^4} + \frac{\partial \tilde{u}}{\partial \xi} \frac{\partial^2 \tilde{u}}{\partial \xi^2} + \frac{3 c_3^* a}{2 c_2^*} \left(\frac{\partial \tilde{u}}{\partial \xi} \right)^2 \frac{\partial^2 \tilde{u}}{\partial \xi^2} = 0. \quad (\text{R.9})$$

Taking $v = \partial \tilde{u} / \partial \xi$ results in the KdV equation:

$$\frac{\partial v}{\partial \tau} + v \frac{\partial v}{\partial \xi} + \beta v^2 \frac{\partial v}{\partial \xi} + \Gamma \frac{\partial^2 v}{\partial \xi^2} + \delta^2 \frac{\partial^3 v}{\partial \xi^3} = 0, \quad (\text{R.10})$$

where $\beta = 3c_3^*a/2c_2^* = 3c_3/2c_2$, $\Gamma = \zeta c_1^*/c_2^* = \zeta a/c_2$, and $\delta^2 = c_1^*a/24c_2^* = a^2/24c_2$ are constants representing the nonlinear advection (soliton), dissipation (or diffusion), and dispersion effects, respectively.

However, as the reviewer correctly pointed out, the continuum limit assumes that the displacements $x_i(t)$ vary slowly along the chain, with the differences between neighboring displacements $x_{i+1} - x_i$ being small. Additionally, the lattice spacing a must be small, and the total length of the material is assumed to be infinite to justify the Long-Wave Approximation.

Per the reviewer's suggestion, we have included a reference to the problem in the introduction and discussion, as follows.

Introduction addition: We note that this chain is a variant of the celebrated Fermi-Pasta-Ulam-Tsingou (FPUT) model [*Fermi, E., Pasta, P., Ulam, S. and Tsingou, M. Studies of the nonlinear problems. Tech. Rep., Los Alamos National Laboratory (LANL), Los Alamos, NM (United States) (1955)*], whose initial study is widely regarded as responsible for the birth of experimental mathematics [*Porter, M. A., Zabusky, N. J., Hu, B. and Campbell, D. K. Fermi, pasta, ulam and the birth of experimental mathematics: numerical experiment that enrico fermi, john pasta, and stanislaw ulam reported 54 years ago continues to inspire discovery. Am. Sci. 97, 214–221 (2009)*]. The FPUT system has also been shown equatable to nonlinear continuum models such as the Kortweg-de Vries (KdV) [*Dauxois, T. and Peyrard, M. Physics of Solitons (Cambridge University Press, 2006)*] and the nonlinear Schrödinger equation (for the case of envelope solitons in diatomic systems [*Huang, G. and Hu, B. Asymmetric gap soliton modes in diatomic lattices with cubic and quartic nonlinearity. Phys. Rev. B 57, 5746 (1998)*]), and formed the foundation for extensions into higher dimensions [*Scott, A. Encyclopedia of nonlinear science (Routledge, 2006)*].”

Discussion addition (also included in the response to the prior question): “While we have demonstrated our design method on a relatively simple (1D, elastic regime, large, few unit cell) system, and extracted several new fundamental insights for our primary case study of peak transmitted kinetic energy minimization, we suggest this sets a foundation for future research and technological development. Most directly ... Similarly immediate, given the flexibility of a designer nonlinear spring and the incorporation into a lumped-mass context shown herein, we suggest this enables the possibility for rapid experimental investigation of nonlinear dynamical phenomena only seen thus far in simulation and theory...”

In each of the above cases, in addition to opening new application possibilities, we believe that this method has the potential to help answer fundamental questions concerning the interplay of nonlinearity with these varied mechanisms, and most generally the spatiotemporal partition of energy in nonlinear dynamical systems [*Fermi, E., Pasta, P., Ulam, S. and Tsingou, M. Studies of the nonlinear problems. Tech. Rep., Los Alamos National Laboratory (LANL), Los Alamos, NM (United States) (1955)*]. Such nonlinear dynamical questions increase in complexity greatly, when considering the influence of disorder [*Pikovsky, A. and Shepelyansky, D. Destruction of anderson localization by a weak nonlinearity. Phys. review letters 100, 094101 (2008)*], including in the form of designed defect placement or spatially extended heterogeneity. It also raises fundamental questions, and via this method a possible new avenue for answering, in the form of

design of nonlinear systems in the context of stimuli sensitivity. For instance, in our case of peak transmitted kinetic energy minimization, we observed (as common for many nonlinear systems) large sensitivity of the kinetic energy transmission metric to small changes in nonlinearity and impact conditions. This indicates future possibilities where computational design can incorporate stimuli sensitivity and parameter uncertainty into the design objective. Further, one can also imagine the potential to leverage the tools developed in the nonlinear dynamical systems community. Instead of optimizing for features in the time domain response of the system, one may optimize in the context of bifurcation diagrams [Boechler, N. et al. *Bifurcation-based acoustic switching and rectification. Nat. materials* 10,665–668 (2011)] or phase space [Strogatz, S. H. *Nonlinear dynamics and chaos: with applications to physics, biology, chemistry, and engineering (CRC press, 2018)*], where in the case of the former, new modes emerge and disappear with variation of input parameters or change stability, and in the latter, nonlinear dynamical objects such as limit cycles and strange attractors are formed. Noting the capacity for systems such as ours to be described via partial differential equations and continuum approximations (most directly, such as the KdV description of the FPUT system [Dauxois, T. and Peyrard, M. *Physics of Solitons (Cambridge University Press, 2006)*.]), analysis tools developed for these limits [Scott, A. *Encyclopedia of nonlinear science (Routledge, 2006)*] may be incorporated into optimization objectives to improve the ability for the computer to meaningfully traverse and efficiently search the design space. Considering all these possibilities together, results in a rich array of possible future extensions.”

2) I got a little confused on what the system parameters that have to chosen are and what the solution measures are; it would help the reader, if around line 80, the authors simply stated what the control parameters are and what they will measure.

We thank the reviewer for their suggestion. We have revised the introductory paragraph for the noted section, to clarify the control parameters and what we will measure, as follows:

“In this section, we describe the identification of an optimal nonlinear constitutive law for the case study of minimizing transmitted peak kinetic energy in response to an impact. Specifically, as shown in Fig. 1A, we simulate the impact of a rigid, variable mass and velocity rigid “impactor” incident on the left end of the chain. The model also incorporates a contact spring designed to facilitate the smooth contact and controlled release of the impactor during initial impact and rebound, respectively. We first consider a single impact condition ($M/M_0 = 0.05$ and $V/V_0 = 1$), where M_0 is half the mass of the chain, V_0 is the linear sound speed, and M and V are the dimensional impactor mass and velocity, respectively. The material is composed of 20 particles and $\zeta = 0.01$. Our control parameters (*i.e.*, design variables for optimization) are the nonlinear coefficients of the springs c_2 and c_3 , which we vary in the aim of minimize the maximum kinetic energy experienced at the end of the material (KE_{non}) normalized by that of a linear system (KE_{lin}) which has all of the same properties except $c_2 = c_3 = 0$ (we hereafter refer to this ratio as the “KE ratio”, where smaller numbers equate to better performance of the nonlinear system).”

3) By the way, why only a cubic polynomial? I’m not sure this was sufficiently motivated, perhaps the authors could expand on this a little more? I mean presumably the order has to be odd to get bi-stability, and a cubic is the simplest way to achieve this, but perhaps the authors could expand on this a bit more.

The reviewer is correct. We chose a cubic polynomial because it captured a broad qualitative range of nonlinearities (*e.g.*, hardening, softening, non-monotonic, and symmetric and asymmetric behavior). Further, we found that inclusion of up to fifth order polynomial terms resulted in minimal performance improvements compared to the third order representation for our primary case study of peak transmitted kinetic energy minimization.

We have clarified this point in the revised manuscript, as follows: “The nonlinear spring consists of an up-to-third-order polynomial, where the linear stiffness remains fixed at c_1^* . The non-dimensional nonlinear spring force is expressed as

$$f(\Delta x) = \Delta x + c_2(\Delta x)^2 + c_3(\Delta x)^3, \quad (\text{R.11})$$

where $\Delta x = \Delta x^*/a$ is the dimensionless spring stretch (with Δx^* the spring stretch defined such that elongation is positive), and c_2 and c_3 are dimensionless nonlinear coefficients of the second and third order-terms (with $c_n = c_n^* a^{n-1}/c_1^*$). We choose to describe the nonlinear springs as up-to-third-order polynomials due to the flexibility of this representation, namely, the ease with which they can represent a wide qualitative range of nonlinearities and the ease which polynomials lend to accurate dynamical simulation (as opposed to non-differentiable, *e.g.*, piecewise continuous functions). Along these lines, we found that inclusion of up to fifth order polynomial terms resulted in minimal performance improvements compared to the third order representation for our primary case study of peak transmitted kinetic energy minimization (see Supplementary Information Note 2).”

“SI—Note 2: Fifth order springs

Here we consider a nonlinear springs with a fifth order polynomial to govern its force-displacement relation, $f(\Delta x) = \Delta x + c_2(\Delta x)^2 + c_3(\Delta x)^3 + c_4(\Delta x)^4 + c_5(\Delta x)^5$. Considering deformations in the range $\Delta x \in [0, 1]$, the positive strain energy condition becomes

$$P(\Delta x) = \int_0^1 f(\Delta x) d\Delta x > 0, \quad (\text{R.12})$$

which leads to a constraint on the polynomial coefficients

$$30 + 20c_2 + 15c_3 + 12c_4 + 10c_5 > 0. \quad (\text{R.13})$$

The material system is simulated via DEM with $N = 20$ particles in the chain, with damping $\zeta = 0.01$ and with the impact conditions $M/M_0 = 0.05$ and $V/V_0 = 1$. For the optimization, the objective is chosen as the peak kinetic energy at the last particle, normalized by the linear response, *i.e.*, $\max(KE_{non})/\max(KE_{lin})$. A gradient based optimizer is used to minimize this objective, while constraining the design space to satisfy the positive energy condition, Eq. R.13. The best nonlinearity obtained from the optimization is $f(\Delta x) = \Delta x - 4.71\Delta x^2 + 1.97\Delta x^3 + 17.74\Delta x^4 - 15.05\Delta x^5$, which results in a performance of $\max(KE_{non})/\max(KE_{lin}) = 3.8\%$. The resultant optimized spring and the corresponding spatiotemporal KE response are shown in Fig. R10. The fifth order optimized springs display qualitative and quantitative similarity to the cubic polynomial springs discussed in the main text within the current deformation range; both having snap-through mechanisms and achieving comparable dynamic performances, thus demonstrating the adequacy of cubic polynomials for KE minimization.”

Figure R10: Results of optimized 5th order springs. (a) Spatiotemporal kinetic energy response. (b) Force-displacement relations of springs.

4) Reading through the article, I think I needed a bit more of an explanation as to how the geometry could be changed of the unit cell to mimic the nonlinearity. I am a mathematician and often struggle with experimental figures (which is my fault I know) but I would've loved a simpler explanation and accompanying figure to understand how the physical system can be adapted; this is Nature Comm. and should be as accessible as possible. I know there was an explanation in the supplementary material, but, in my humble opinion, I think the authors missed a trick here by not being super-transparent about this in the main text.

We thank the reviewer for this suggestion, and have added additional panels to the experimental setup figure (now Fig. 5 in the main text, Fig. R11 below in the response letter) to further clarify the process by which the geometry can be manipulated to achieve various types of nonlinearities (panel A). A variety of geometry-driven nonlinearity mechanisms have been included, and the optimization process has been highlighted (panels B and C). The latter includes a flow-chart-like diagram that shows how the optimized design translates into the physical experimental realization.

In addition, the following text has been added to the manuscript: “The constitutive response of a spring element is directly tied to both its constituent material and its geometry. Given that the response of the underlying constituent material is accounted for (*e.g.*, neo-Hookean or Saint Venant-Kirchhoff), the geometry can be designed to tailor the effective constitutive response of the spring. Several example known mechanisms for achieving various broad classes of nonlinearity are highlighted in Fig. R11A. This is accomplished through coarse geometry adjustment—that is, only altering the angle, length, and thickness of the beam. The underlying mechanisms can be intuitively thought of as follows: when the thin beam-like spring element undergoes large deformation, if it deforms into a state where it is being stretched axially, it will stiffen, while if it is deformed to where it is being loaded transversely, it will soften. If in the latter case it is deformed such that it undergoes axial compression against the boundaries, it tends towards negative stiffness (*e.g.*, snap-through or bistability). While these coarse adjustments are adequate to achieve broad types of nonlinear behavior, they are inadequate to achieve the type of precision of nonlinear response needed in dynamical settings—*e.g.*, the subtle difference between the optimal spring response (red line) and the underperforming (bad) spring response (dashed-dotted blue line) shown in Fig. R5.”

Figure R11: **Designing unit cells that leverage geometric nonlinearity to achieve DEM-identified desired effective constitutive laws, and experimental realization of the spring and chain.** A) Example mechanisms which can be used to achieve various nonlinearities with geometry alone are shown, including the following nonlinearities: i) stiffening; ii) softening; iii) soft to stiff; iv) snap-through (but not bistable) v) fully bistable. The plot on the right shows the nonlinear responses of these five example mechanisms (with the stiff to soft, snap-through, and bistable curves linearly scaled to be on the same scale as the thicker mechanisms). B-E) Optimized spring design and chain realization for the case of minimizing peak transmitted kinetic energy. B) Results of the shape optimization, with the initial condition and optimized design shown. C) The fabricated polycarbonate unit cell, consisting of four optimized springs (as seen in (B)) and a rigid frame to impose boundary conditions. D) The quasi-static test of the unit cell shown in (C), compared against the target behavior as well as the behavior of a single spring with perfectly imposed boundary conditions as simulated via COMSOL FEM simulation. E) The full chain of 20 unit cells, hung from a frame. The chain is clamped to the left of the leftmost unit cell, imposing a zero displacement boundary condition. The impact occurs at the right end of the chain. The unit cell and spring length scales are labeled.

5) **The discussion:** This is where I think the paper needed the most work; the future work was described as simply changing the objective function in the optimization problem which I think is rather putting this research down a bit; surely there must be some more important applications which I don't think the authors spelt out clearly enough.

We thank the reviewer for their insight. The discussion has been drastically expanded to provide further context and clarity regarding future possibilities, implications, and applications. The text now reads as follows (also included as response to the prior questions):

“As suggested by the physically large size of our experimentally realized system, we note that this class of geometrically-nonlinear-microstructure-enabled wave transforming materials demands a particularly challenging set of manufacturing requirements. This is due to a particularly large scale separation between the system size and the smallest features. For instance, the optimized design shown in Fig. R11B contains very thin hinges at the top and bottom of the central beam, particularly compared to the size of the unit cell. Given that the wave-dominated behavior shown herein requires many unit cells, we have a

situation with three distinct separated length scales (smallest feature, unit cell, system). While this length scale separation is in some sense shared by other metamaterial and lattice structures [Cummer, S. A. et al. *Controlling sound with acoustic metamaterials. Nat. Rev. Mater.* 1, 1–13 (2016)., Jiao, P. et al. *Mechanical metamaterials and beyond. Nat. communications* 14, 6004 (2023)], it is amplified for our case. For instance, at the unit cell level, reduction in this disparity is possible, however it comes at the expense of limiting the range of nonlinearity achievable (*i.e.* one often needs long, thin, deformable structures to achieve large nonlinear response within elastic regimes). Future incorporation of plasticity, phase transformation, or contact may address this issue. Further, recent works have shown the existence of a nonlinear dynamic “size effect” where a critical, minimal number of unit cells has been shown to yield enhanced performance [Xiu, H. et al. *Minimizing finite viscosity enhances relative kinetic energy absorption in bistable mechanical metamaterials but only with sufficiently fine discretization: a nonlinear dynamical size effect. arXiv preprint arXiv:2410.02090 (2024)*], in addition to the longer system size compared to the characteristic wavelengths giving more time and space for traveling pulses to evolve. Indeed, relatively coarse parameter sweeps of the cubic polynomial coefficients for the peak transmitted kinetic energy minimization problem for more highly discretized materials (*e.g.*, 100 unit cells, see Supplementary Information Note 7), suggest the potential for over two orders of magnitude improvement in kinetic energy transmission via the use of a nonlinear material compared to a comparative linear material. All of these length scale separation manufacturing challenges would be further exacerbated in 3D, with the addition of loss of the mechanism for tuning the linear stiffness via out-of-plane structure depth. However, we note that recent advances in 3D printing have seen large jumps in the system to smallest feature size ratio versus manufacturing speed [Kiefer, P. et al. *A multi-photon (7× 7)-focus 3d laser printer based on a 3d-printed diffractive optical element and a 3d-printed multi-lens array. Light. Adv. Manuf.* 4, 28–41 (2024)]. Self-assembly-based manufacturing is a further tantalizing possibility [Jin, H. and Espinosa, H. D. *Mechanical metamaterials fabricated from self-assembly: A perspective. J. Appl. Mech.* 91 (2024)].

While we have demonstrated our design method on a relatively simple (1D, elastic regime, large, few unit cell) system, and extracted several new fundamental insights for our primary case study of peak transmitted kinetic energy minimization, we suggest this sets a foundation for future research and technological development. Most directly, within the primary case study of impact mitigation, a near term question of future interest would be how such optimal nonlinearities would change for a different metric such as maximum transmitted force or peak tensile stress anywhere in the material. Similarly immediate, given the flexibility of a designer nonlinear spring and the incorporation into a lumped-mass context shown herein, we suggest this enables the possibility for rapid experimental investigation of nonlinear dynamical phenomena only seen thus far in simulation and theory. Less directly, one can extend the bilevel optimization to 3D and to more complex constituent material models. The extension to 3D yields particularly non-trivial challenges, due to the second-order tensorial nature of solid mechanical systems, including with the potential for shear-to-longitudinal mode conversion in linear [Graff, K. F. *Wave motion in elastic solids (Courier Corporation, 2012)*], and not-to-mention nonlinear [Wallen, S. P. and Boechler, N. *Shear to longitudinal mode conversion via second harmonic generation in a two-dimensional microscale granular crystal. Wave motion* 68, 22–30 (2017)], dynamical regimes. With regards to optimization of unit cells in 3D for desired quasi-static nonlinear mechanical response, some initial steps have recently been made for select multiaxial loading states [Zheng, L. et al., *Hypercan: Hypernetwork-driven deep parameterized constitutive models for metamaterials. Extrem. Mech. Lett.* 72, 102243 (2024)]]. Extension of this method to less scale separated dynamical (no lumped masses) would enable more mass efficient structures, but would likely require either some degree of homogenization or larger, high-performance computing capability. The incorporation of irreversibility of the constituent material model, such as plasticity, would open enhanced energy dissipation mechanisms and application to high strain-rate applications [Meyers, M. A. *Dynamic behavior of materials (John Wiley and Sons, 1994)*]. Similarly, the incorporation of activity or stimuli-responsivity [Xia, X. et al. *Responsive materials architected in space and time. Nat. Rev. Mater.*

7, 683–701 (2022)] in the constituent material model would open applications in shape morphing, mechanical computing, and the capacity for materials that autonomously respond and conduct directed work [Jiao, P. et al. *Mechanical metamaterials and beyond. Nat. communications* 14, 6004 (2023)]. The use of shape memory materials [Xia, X. et al. *Responsive materials architected in space and time. Nat. Rev. Mater.* 7, 683–701 (2022)] offers potential in each of these capacities, via targeted shape change, potential resetting capability, and enhanced energy absorption. Incorporation of multiphysics coupling into the material model may similarly open up new possibilities in fields such as optomechanics [Aspelmeyer, M. et al. *Cavity optomechanics. Rev. Mod. Phys.* 86, 1391–1452 (2014)].

In each of the above cases, in addition to opening new application possibilities, we believe that this method has the potential to help answer fundamental questions concerning the interplay of nonlinearity with these varied mechanisms, and most generally the spatiotemporal partition of energy in nonlinear dynamical systems [Fermi, E., Pasta, P., Ulam, S. and Tsingou, M. *Studies of the nonlinear problems. Tech. Rep., Los Alamos National Laboratory (LANL), Los Alamos, NM (United States) (1955)*]. Such nonlinear dynamical questions increase in complexity greatly, when considering the influence of disorder [Pikovskiy, A. and Shepelyansky, D. *Destruction of anderson localization by a weak nonlinearity. Phys. review letters* 100, 094101 (2008)], including in the form of designed defect placement or spatially extended heterogeneity. It also raises fundamental questions, and via this method a possible new avenue for answering, in the form of design of nonlinear systems in the context of stimuli sensitivity. For instance, in our case of peak transmitted kinetic energy minimization, we observed (as common for many nonlinear systems) large sensitivity of the kinetic energy transmission metric to small changes in nonlinearity and impact conditions. This indicates future possibilities where computational design can incorporate stimuli sensitivity and parameter uncertainty into the design objective. Further, one can also imagine the potential to leverage the tools developed in the nonlinear dynamical systems community. Instead of optimizing for features in the time domain response of the system, one may optimize in the context of bifurcation diagrams [Boechler, N. et al. *Bifurcation-based acoustic switching and rectification. Nat. materials* 10,665–668 (2011)] or phase space [Strogatz, S. H. *Nonlinear dynamics and chaos: with applications to physics, biology, chemistry, and engineering (CRC press, 2018)*], where in the case of the former, new modes emerge and disappear with variation of input parameters or change stability, and in the latter, nonlinear dynamical objects such as limit cycles and strange attractors are formed. Noting the capacity for systems such as ours to be described via partial differential equations and continuum approximations (most directly, such as the KdV description of the FPUT system [Dauxois, T. and Peyrard, M. *Physics of Solitons (Cambridge University Press, 2006)*]), analysis tools developed for these limits [Scott, A. *Encyclopedia of nonlinear science (Routledge, 2006)*] may be incorporated into optimization objectives to improve the ability for the computer to meaningfully traverse and efficiently search the design space. Considering all these possibilities together, results in a rich array of possible future extensions.”

Responses to Reviewer 3:

The work shows an experimental realization of a material with tailored nonlinear constitutive responses. The system is obtained via a shape optimization inverse design and validated experimentally with good agreement. The methodology shown in the work allows for on-demand quasi-static nonlinearity which is of relevance for the community of mechanics. The article is well written, covers a broad audience and reaches the standards of the journal, however there are several points to be revised and improved. I suggest publishing the article once these points will be solved.

We thank the reviewer for their positive assessment of our work, and their constructive feedback.

1) **The general expression of $f(\Delta x)$ is provided in the Note 3 of the supplementary. I think this expression is important enough to mention it in the main article. In fact, the optimization is based on this. Moreover, the Physical meaning of the terms in Eq. (1) may be explained for the clarity of the reading. In a general way, I think that the discussion of Note 3 is important enough to be synthesized in the main text. Figs. 2A, 2B and 2C are very difficult to follow without this supplementary material. In my opinion the main text is not enough to understand the results. I encourage the authors to introduce a summary of the Note 3 in the main article.**

We thank the Reviewer for this suggestion. In the revised manuscript, we have added: i) An additional equation which shows $f(\Delta x)$ with a description of the physical meaning of the variables; and ii) A summary of note 3 in the main text (highlighting that what was “note 3” in the original manuscript is now “note 5”). The revised sections are included as follows.

i) “The nonlinear spring consists of an up-to-third-order polynomial, where the linear stiffness remains fixed at c_1^* . The non-dimensional nonlinear spring force is expressed as

$$f(\Delta x) = \Delta x + c_2(\Delta x)^2 + c_3(\Delta x)^3, \quad (\text{R.14})$$

where $\Delta x = \Delta x^*/a$ is the dimensionless spring stretch (with Δx^* the spring stretch defined such that elongation is positive), and c_2 and c_3 are dimensionless nonlinear coefficients of the second and third order-terms (with $c_n = c_n^* a^{n-1}/c_1^*$). We choose to describe the nonlinear springs as up-to-third-order polynomials due to the flexibility of this representation, namely, the ease with which they can represent a wide qualitative range of nonlinearities and the ease which polynomials lend to accurate dynamical simulation (as opposed to non-differentiable, *e.g.*, piecewise continuous functions). Along these lines, we found that inclusion of up to fifth order polynomial terms resulted in minimal performance improvements compared to the third order representation for our primary case study of peak transmitted kinetic energy minimization (see Supplementary Information Note 2).”

ii) “Before searching for optimal nonlinear constitutive responses with our DEM, we set several bounds. First, for simplicity, we set the tension response to purely linear. In Supplementary Information Note 4, we show that the inclusion of nonlinearity in tension has little effect on the identified optima, which is as expected, due to the compressive nature of the impact event simulated (the identified optimum has a maximum compressive strain over five times greater than the maximum tensile strain). Second, we confine the unit cell strain to 1 in compression, and set $c_3 > 0$ for simplicity. Third, we restrict our search range for nonlinear coefficients c_2 and c_3 to ensure positive strain energy throughout the entire compression range. We note that keeping the linear stiffness constant, we exclude essential nonlinearities [Nesterenko, V. Dynamics of heterogeneous materials (Springer Science & Business Media, 2013)]. By examining the polynomial’s properties within this range, we classify the quasi-static response into three distinct zones, “bistability”, “monotonic increase”, and “local maximum”, as shown in Fig. R5A. Bistability (magenta area) denotes the existence of both a local maximum and minimum other than the boundaries (the local minimum does not need to fall below zero). Monotonic increase (blue area) denotes the absence of extrema. Local maximum (green area) signifies the presence of a local maximum (no local minimum existed) within the range of the length of one unit cell. For a monotonic increase of $f(\Delta x)$, the condition $f'(\Delta x) \geq 0$ must be satisfied, or the local maximum of $f(\Delta x)$ should occur at $\Delta x \geq 1$. To ensure bistability, both local maximum and minimum of $f(\Delta x)$ are set to be located within $\Delta x \in (0, 1)$. For a local maximum property to be exhibited, we must have $0 < \Delta x_1 < 1$ and $\Delta x_2 > 1$, where Δx_1 and Δx_2 are the roots of $f'(\Delta x) = 0$. More details concerning these zones are given in Supplementary Information Note 5.”

2) In Eq. (2) Ω is not defined. It is defined later but not here.

We thank the Reviewer for catching this oversight. In the revised manuscript, we have defined Ω immediately after Eq. (2), as shown below:

The exact form of the objective function is

$$\min_{\Omega} \sum_{i=2}^n \left(\frac{c_i T_1}{c_1 T_i} - 1 \right)^2, \quad (\text{R.15})$$

where Ω is the design domain (*i.e.*, the range of values that all the design variables can take), c_i and c_1 represent the current polynomial coefficients (nonlinear and linear, respectively), T_i and T_1 represent the target coefficients (nonlinear and linear, respectively), and n represents the total number of nonlinear polynomial coefficients.

3) Dashed line in Fig. 5E is not defined.

We thank the Reviewer for pointing out this oversight. In the revised manuscript, we have defined the dashed line in Fig. 5E in the figure caption (now Fig. 6G in the revised manuscript), as: “A value greater than 1 (denoted by the horizontal dashed black line) indicates superior performance of the nonlinear chain as compared to the linear.”

4) A short discussion about the sensibility of the shape optimization to the properties of the impactor. Apparently, there is a huge sensitivity to small variations of the impactor.

We thank the Reviewer for this suggestion. In the revised manuscript, we have added the following additions (in blue) to our discussion of the sensitivity to impact conditions in the main text. We note also SI Note 11 (copied here for convenience) that discusses this observed sensitivity.

“A point of particular note, is that although the targeted conditions show excellent predicted performance, the behavior can be sensitive to small variations in impactor mass and velocity, as seen in the simulation data of Fig. R8H. The variability of the mass-velocity space is immediately apparent, with several very small regions of excellent performance (low KE ratios) surrounded by oscillating regions of lower performance (relatively higher KE ratios), and even several points of poor performance (KE ratio > 1). While the region targeted in this work sought a region with relatively low drops in performance (relative to other regions explored via simulation), there still exists varied performance impact conditions nearby. In order to more quantitatively describe this sensitivity, in Supplementary Information Note 11, we calculate the gradient of Fig. R8H, which shows $|\partial(\log_{10}(KEratio))/\partial M|$ can reach near 1 g^{-1} and $|\partial(\log_{10}(KEratio))/\partial V|$ can reach up to 40 s/m . This means that, for the most sensitive regions of the impact conditions landscape, a change in 1 g of impactor mass can result in an up to $\sim 6.7\times$ change in KE ratio, or a 0.01 m/s change in impactor velocity can result in an up to $\sim 2.5\times$ change in KE ratio. However, as with the case of the sensitivity to the nonlinear stiffness parameters, near our chosen optimum, the sensitivity is significantly lower, with $\partial(\log_{10}(KEratio))/\partial M = 0.16 \text{ g}^{-1}$ and $\partial(\log_{10}(KEratio))/\partial V = -7.8 \text{ s/m}$.

This raises the question of why is this nonlinearity so sensitive to impactor conditions. Nonlinear systems in general are known to be sensitive to small changes in system parameters, including initial and boundary conditions, particularly near points of bifurcation or instability [Strogatz, S. H. *Nonlinear dynamics and chaos: with applications to physics, biology, chemistry, and engineering* (CRC press, 2018)] (*e.g.*, the butterfly effect [Scott, A. *Encyclopedia of nonlinear science* (Routledge, 2006)]). In our case, we suggest this sensitivity is due to two factors: i) The bistability present in the identified optima studied

in experiment, and ii) Our choice of performance figure of merit. Considering the effects stemming from bistability, in Supplementary Information Note 11 we show that large, cyclical, changes in performance are seen with small changes in impactor velocity, where these changes correlate with the arrest of the initially generated solitary wave (as seen in Fig. 1B, Fig. R12D and E, and Fig. 6A and B) moving one unit cell closer to the boundary opposite the impactor. Further, this solitary wave takes the form of snapping and unsnapping of each unit cell in sequence until, generally (as can be seen in Supplementary Information Note 11 Fig. S25), a unit cell remains snapped shut, which corresponds to the point of solitary wave arrest. This is consistent with the idea that with increasing impactor energy, the final unit cell that snaps shut locks in more and more energy until it reaches a critical point, at which it suddenly unsnaps, releasing energy back into the system in the form of kinetic energy. This effect is further amplified, as per our second suggested factor, because we choose peak kinetic energy at the last particle in the chain. We suggest that some of the identified sharp changes in performance are due to the effect seen in Fig. R12D and E, wherein the “best” case of Fig. R12D shows solitary wave arrest just before reaching the boundary, whereas the slightly different nonlinearity of the “bad” case of Fig. R12E allows the solitary wave to interact with the boundary. We expect similar phenomena occur due to the cyclic advance of solitary wave’s arrested position with increased impactor velocity. In addition to these factors, we also expect a contribution from wave interference, where small bits of reflected energy can push a unit cell above or below snapping and unsnapping thresholds.

Figure R12: **Identification of optimal nonlinear constitutive response via DEM simulation for a single impact condition for the case of peak transmitted kinetic energy minimization.** (A) Feasible solutions of nonlinear spring coefficients c_2 and c_3 . The black line represents $c_2 = -\sqrt{3}c_3$, the red line indicates $c_2 = -(1 + 3c_3)/2$, the blue line is $c_2 = -3c_3$, and the green line is the zero strain energy throughout the whole range, $c_2 = -3/2 - 3c_3/4$. (B) Non-dimensional force-extension relationship of the best performing nonlinear spring ($f(\Delta x) = \Delta x + 5.88\Delta x^2 + 9.65\Delta x^3$) along with an example of a nearby underperforming (bad) nonlinear spring of $f(\Delta x) = \Delta x - 6.6\Delta x^2 + 11\Delta x^3$ (blue) and an energy locking bistable spring of $f(\Delta x) = \Delta x - 5.26\Delta x^2 + 6.75\Delta x^3$ (magenta). These three cases result in KE ratios of 0.0398 (best), 1.2257 (bad), and 0.1448 (energy locking bistable). (C) Ratio of maximum kinetic energy of the nonlinear spring to the one of a linear spring at the last particle as a function of nonlinear spring coefficients for the impact condition of $M/M_0 = 0.05$ and $V/V_0 = 1$, with $\zeta = 0.01$. The lines from (A) are overlaid, the star marker denotes the point of best performance, the triangle indicates the nearby case, and the square represents the bistable case. Normalized kinetic energy of the (D) best performing nonlinear, (E) underperforming (bad) nonlinear, and (F) linear material.

As pertains particularly to the experimental results, we note that the impactor velocities obtained herein were not precise (ranging from 1.36-1.41 m/s, see Supplementary Information Note 9), and this, coupled with the aforementioned sensitivity provides an insight into variations in chain performance. We believe the proximity of regions of poor performance to regions of good performance (*e.g.*, Fig. R8H) motivates a consideration of nearby conditions in future work. For instance, we expect there are application scenarios in which a region of reduced sensitivity to changes in stimuli may be desirable at the expense of slightly lowered performance. Despite the sensitivity of the system, however, per Fig. R8G the kinetic energy ratio (KE_{lin}/KE_{non}) remains greater than 1 (superior performance of the nonlinear chain compared to the linear) for both simulation and experiment once a critical number of unit cells has passed (unit cell 11 for the simulation, unit cell 10 in experiment).”

“Note 11: Sensitivity to system and impactor properties, and sharp performance changes upon unit cell snapping **for the case of peak transmitted kinetic energy minimization.**

In Fig. R13(a,b), we show the sensitivity of the \log_{10} of the data in the Fig. 6H in the main text (KE ratio) with respect the impactor conditions M and V , respectively.

We next demonstrate the sharp changes in performance (maximum kinetic energy experienced at the end

Figure R13: Performance gradients. Sensitivity of the \log_{10} of the data in Fig. 6H in the main text (KE ratio) with respect to the impactor conditions (c) M and (d) V ($\partial(\log_{10}(KE \text{ ratio}))/\partial M$ and $\partial(\log_{10}(KE \text{ ratio}))/\partial V$, respectively).

of the linear chain, divided by the maximum kinetic energy experienced at the end of the nonlinear chain) when an additional unit cell in the chain snaps. Here we consider a slightly different, but still bistable, nonlinear spring from the main text. The parameters for this system are $f(\Delta x) = \Delta x + 87\Delta x^2 + 1778\Delta x^3$, where the linear stiffness is set as $c_1^* = 1.0454$ kN/m, the impactor mass $M_0 = 40$ g, and the length of a unit cell $a = 125$ mm. In Fig. R14, we see very large, periodic jumps in performance (a factor of greater than 10 times) as the impactor velocity is increased even by very small amounts of $\sim 3\%$ (~ 0.03 m/s). Here we consider a shorter simulation duration, just over the one-way transit time of a pulse through the material.

Several of the simulations underlying the parameter sweep of Fig. R14 can be seen in Fig. R15, for impactor velocities corresponding to several sequential maximum amplitude points in Fig. R14 ($V = \{1.354, 1.406, 1.454, 1.497\}$ m/s). It can be seen in Fig. R15, that each of these high amplitude points sequentially correspond to one later unit cell in the chain snapping and terminating the motion of the main initial pulse propagating through the system.”

Figure R14: The maximum kinetic energy (linear/nonlinear) versus impactor velocity at the last unit cell using a new set of nonlinear spring coefficients. The new nonlinear mechanical response has a similar behavior to the one used in the experiment with the form of $f(\Delta x) = \Delta x + 87\Delta x^2 + 1778\Delta x^3$. The linear stiffness is set as $c_1^* = 1.0454$ kN/m, the impactor mass is 40 g, and the length of a unit cell is 125 mm.

Figure R15: Spatiotemporal evolution of strain in the nonlinear chain for different impactor velocities corresponding to several of the highest amplitude KE ratio peaks in Fig. R14. (a) $V = 1.354$ m/s, (b) $V = 1.406$ m/s, (c) $V = 1.454$ m/s, and (d) $V = 1.497$ m/s.